# Spatial single-cell mass spectrometry defines zonation of the hepatocyte proteome

Florian A. Rosenberger [1], Marvin Thielert [1], Maximilian T. Strauss [2], Lisa Schweizer[1], Constantin Ammar[1], Sophia C. Mädler[1], Andreas Metousis[1], Patricia Skowronek [1], Maria Wahle[1], Katherine Madden[1], Janine Gote-Schniering[3], Anna Semenova[3], Herbert B. Schiller[3], Edwin Rodriguez[1], Thierry M. Nordmann[1], Andreas Mund [2] & Matthias Mann [1,2] ✉

Single-cell proteomics by mass spectrometry is emerging as a powerful and unbiased method for the characterization of biological heterogeneity. So far, it has been limited to cultured cells, whereas an expansion of the method to complex tissues would greatly enhance biological insights. Here we describe single-cell Deep Visual Proteomics (scDVP), a technology that integrates high-content imaging, laser microdissection and multiplexed mass spectrometry. scDVP resolves the context-dependent, spatial proteome of murine hepatocytes at a current depth of 1,700 proteins from a cell slice. Half of the proteome was differentially regulated in a spatial manner, with protein levels changing dramatically in proximity to the central vein. We applied machine learning to proteome classes and images, which subsequently inferred the spatial proteome from imaging data alone. scDVP is applicable to healthy and diseased tissues and complements other spatial proteomics and spatial omics technologies.

Mass spectrometry (MS)-based single-cell proteomics (scProteomics) has made tremendous progress within just a few years, and can now quantify more than 1,000 proteins in cultured cells[1–3]. While this trajectory is promising, proteome depth, throughput and lack of spatial context limit biological use. We have recently introduced deep visual proteomics (DVP), a spatial technology that combines imaging, cell segmentation, laser microdissection and MS into a single workflow to investigate complex tissues with various cell types and metabolic niches[4]. DVP overcomes depth and throughput limitations with pooling the required number of cells with similar morphological features and staining patterns to identify statistically and analytically robust cellular phenotypes ('biological fractionation'). By its nature, it depends on prior knowledge of adequate markers of the cells of interest that resolve their heterogeneity. These markers might not be available for all subtypes of cells or those tissues that have rapidly

changing proteome types such as heterogeneous tumors. To address this, we here developed single-cell DVP (scDVP), a complementary approach that extends scProteomics technologies into the intact tissue context.

In this Article, we use scDVP to explore spatial characteristics of hepatocyte subsets in mammalian liver—a highly organized and functionally repetitive tissue, in which the proteome of hepatocytes is determined by paracrine signaling, as well as oxygen and nutrient gradients[5]. These metabolic gradients require distinct functional cell states along the portal vein (PV) to central vein (CV) axis. This phenomenon of liver zonation has been described by single-cell RNA sequencing (scRNAseq) for hepatocytes[6,7], fluorescence-activated cell sorting (FACS) and MS-based proteomics[8], and multiplexed imaging[9]. Despite this long and varied background, the extent of spatial heterogeneity and proteome variation in hepatocyte remains an open question.

[1]Proteomics and Signal Transduction, Max Planck Institute of Biochemistry, Martinsried, Germany. [2]Proteomics Program, Novo Nordisk Foundation Center for Protein Research, Faculty of Health and Medical Sciences, University of Copenhagen, Copenhagen, Denmark. [3]Comprehensive Pneumology Center (CPC) / Institute of Lung Health and Immunity (LHI), Helmholtz Munich; Member of the German Center for Lung Research (DZL), Munich, Germany. ✉e-mail: mmann@biochem.mpg.de

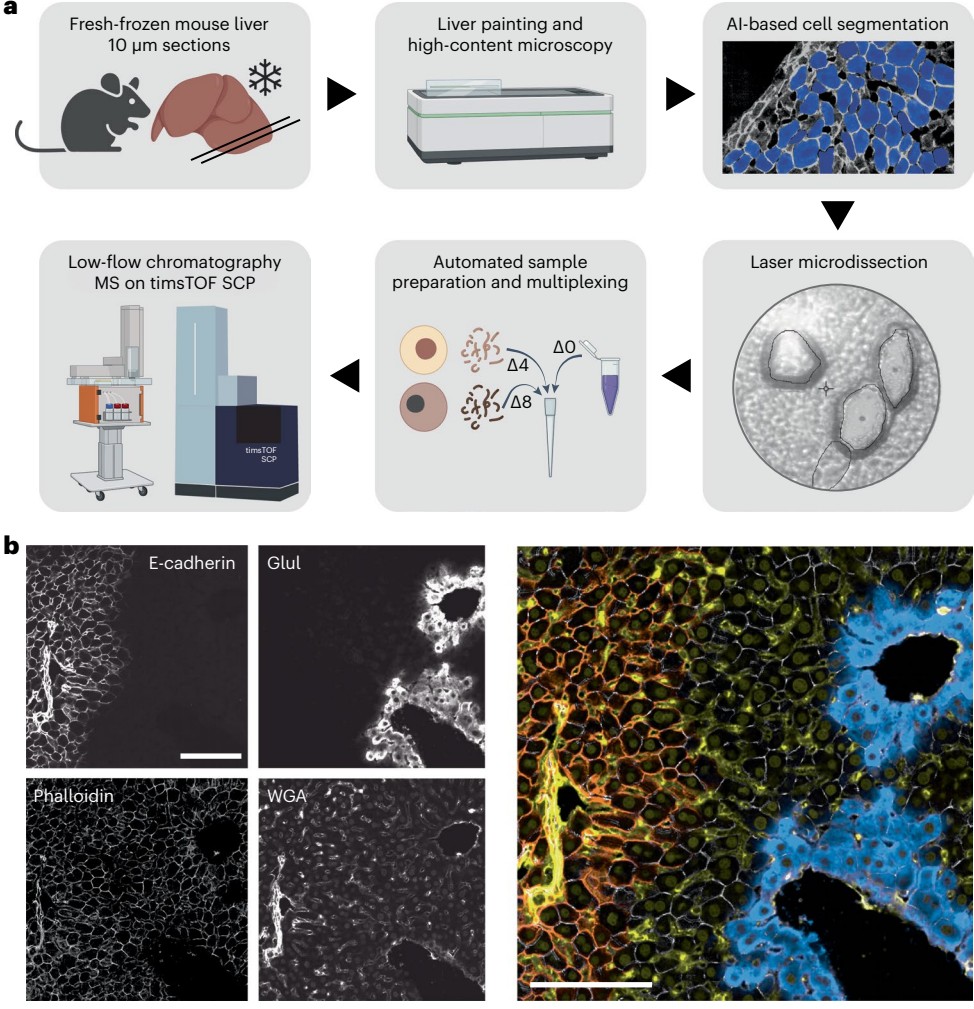

**Fig. 1 | Isolation and characterization of individual hepatocyte shapes in situ. a**, The scDVP workflow comprised embedding of fresh mouse liver tissue, staining and high-content microscopy, AI-guided hepatocyte segmentation, cutting and sorting of cells on a laser microdissection microscope, and peptide preparation with or without dimethyl labeling. The Δ0 channel contains the reference proteome and Δ4 and Δ8 contain two individual samples, which are all analyzed by ultra-high-sensitivity mass spectrometry. Created with BioRender. com. **b**, Liver painting with four stains. Left: E-cadherin marks PV regions, glutamate-ammonia ligase (Glul) surrounds the CV, the cell segmentation marker phalloidin, and the sinusoidal and nuclear counterstain WGA. Right: false color overlay of all channels: orange, E-cadherin; yellow, WGA, gray, phalloidin; turquoise, Glul. Scale bars, 100 μm.

## Results

### Robust isolation and characterization of hepatocyte shapes

To map the proteome of mouse hepatocytes at single-cell resolution, we established a modular and automated workflow aimed at loss-less sample preparation of the initial input cell for injection into the mass spectrometer (Fig. 1a). Mice livers were embedded and immediately frozen after cardiac arrest. We fixed 10 μm sections and stained them with a one-step protocol marking PVs and CVs, the sinusoidal architecture, nuclei and cell membranes (Fig. 1b and Methods). Individual cells were segmented by deep learning as before[4], and the resulting masks transferred to a laser microdissection microscope that automatically excised and collected individual shapes in 384-well plates. Given hepatocyte sizes of 20–30 μm, one shape cut from a 10 μm section corresponds to a third or half of a hepatocyte, or approximately 250 pg of protein input, equivalent to the protein content of one HeLa cell. We automated protein extraction and digestion by reagent addition into the same plate, omitting extra transfer steps, followed by peptide separation on the Evosep system[10] and injection into a trapped ion mobility time-of-flight single-cell proteomics (timsTOF SCP) mass spectrometer (Fig. 1a).

To establish an efficient workflow, we applied our scProteomics protocol[2] and titrated the number of cells required to obtain a robust signal (Extended Data Fig. 1a). To confirm biological ground truth, we performed initial experiments on five adjacent shapes per well (corresponding to about two complete hepatocyte cell masses), cut from randomly chosen locations. With these five shapes, we reached a median depth of 1,235 proteins across 230 samples (Extended Data Fig. 1b). The results confirmed expected liver biology, for instance, by differential expression of the PV marker argininosuccinate lyase (Asl) and central Cytochrom P450 2E1 (Cyp2e1, Extended Data Fig. 1c). Using zonation anchor proteins to arrange all the samples in pseudo-space (Extended Data Fig. 1d), we characterized spatially enriched gene sets along the zonation axis. While the protein sets for electron transport chain and oxidative phosphorylation (OXPHOS) were among processes upregulated in proximity to the PV, biotransformation and oxidations by cytochrome P450 were increased proximal to the CV, providing positive controls for low-input proteomics (Extended Data Fig. 2 and Supplementary Tables S1 and S2).

### Multiplex-DIA drastically increases proteome depth

Encouraged by these spatial results, we next asked whether single shapes alone could produce deep and interpretable proteomic results. To improve sensitivity further, we adopted and optimized elements of

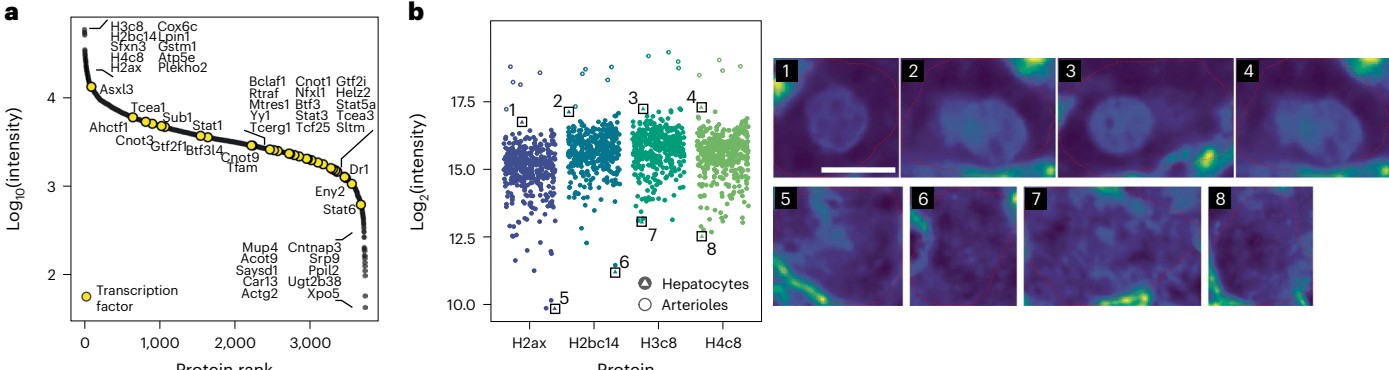

**Fig. 2 | Depth of single-shape proteomes and estimation of the nuclear compartment. a**, Unique proteins quantified in our scDVP workflow with mDIA ranked by signal intensity (two single-shape and a reference proteome channels, 31 min Evosep gradient, 15 cm column at 100 nl min⁻¹, dia-PASEF with optimized window design, library-dependent search in DIA-NN). Names of highest- and lowest-ranking proteins, as well as transcription factors, are indicated. **b**, Left: intensity of the top four histone proteins across all samples, including hepatocytes and quality control arteriole structures. Colors are specific for the indicated histone subunit. Right: WGA stain of cells corresponding to marked data points in the scatter plot. The color scale is signal intensity. Scale bar, 10 μm. Data from three mice were pooled.

our scProteomics workflow[11]. These include addition of the surfactant *n*-dodecyl-β-ᴅ-maltoside (DDM) to maximize peptide recovery[12], lowering the chromatographic flow rate to 100 nl min⁻¹ for increased ionization efficiency[2] ('Whisper gradients' on the Evosep system) and achieving higher chromatographic resolution with zero dead volume columns (IonOpticks)[13]. Most importantly, we added a labeled reference channel for multiplexed data-independent acquisition (mDIA) that decouples identification and quantification[11] (Fig. 1a).

For scDVP, we constructed a dimethyl-labeled bulk liver reference. Our robotic sample preparation setup achieved about 99% labeling efficiency in all three channels (Extended Data Fig. 3a). We co-injected 10 ng of the reference proteome together with the labeled proteomes of two single shapes at a mean size of 600 μm² (Extended Data Fig. 3b). This resulted in a doubling of identified proteins with a median number of 1,712 proteins across three biological replicates and 455 single shapes, at twice the previous throughput (Extended Data Fig. 3c). A maximum of 2,769 proteins were identified in one shape, and 3,738 unique proteins were found across all samples (Fig. 2a and Extended Data Fig. 3c). Four histone components ranked in the top ten, but we also found many transcription factors (Fig. 2a). In contrast, plasma proteins produced in hepatocyte were of medium abundance and hemoglobin subunits were not detected. This suggests little to no contamination from surrounding blood, a common issue in bulk proteomics (Extended Data Fig. 3d). The number of detected proteins correlated logarithmically with the microdissected area (Extended Data Fig. 3e), indicating that scDVP requires the highest possible MS sensitivity. Data completeness across all samples increased with median intensity per protein. Coefficients of variation were less than 50% and strongly depended on cell size and position along the zonation axis, reflecting biological heterogeneity in the data (Extended Data Fig. 3f,g). We hypothesized that the nuclear proportion in the cell slice would correlate with the intensity of these histones. Indeed, shapes with lowest histone intensities did not have any evident nuclear signal, while top intensities were in shapes with large or two nuclei. In addition to this, the intensity of the top four abundant histone proteins was highest in arterioles that we cut as technical control structures, and which are composed of more than one cell and nucleus (Fig. 2b and Extended Data Fig. 3h).

### Single-shape proteomes accurately reflect hepatocyte zonation
To test the biological validity of our proteomics data, we first reduced dimensionality in a principal component analysis (PCA), which revealed that PC1 represented the measured distance of a hepatocyte to PV and CV (Fig. 3a,b). Overlays of known liver zonation markers including Cyp2e1 and argininosuccinate lyase (Asl) showed opposite visual enrichment along PC1 (Extended Data Fig. 4a,b). In contrast, PC2 did not correlate with measured distance or hepatocyte zonation markers but rather with cytoskeletal components (Extended Data Fig. 4c,d). PC2 was also the dimension in which portal arterioles, which we excised as technical controls, separated from hepatocytes (Extended Data Fig. 4e).

We asked whether single-cell resolution provides any benefit over combining adjacent shapes. To this end, we iteratively combined shape information and averaged protein levels of cells with the same relative location along the zonation axis ('pseudo-neighbors'). Starting from the combination of as little as two shapes, PC1 continuously gained importance as measured by interquartile range and variance explained, whereas PC2 and all subsequent components dropped in explanatory value (Extended Data Fig. 5). This demonstrates that single-cell data retains subtle biological differences compared to the excision of larger areas.

On the basis of the distance ratio of PV and CV, we grouped the data into 20 spatial bins—approximately the maximum number of cells along the zonation axis. Analysis of variance (ANOVA) testing across all bins revealed that 49% of all proteins detected in at least half of the samples were significantly different between zones (false discovery rate (FDR) <0.05; Extended Data Fig. 6a and Supplementary Table S2). Zonation was also apparent after spatial sorting at the total proteome level (Fig. 3c and Supplementary Table S3) and for known hepatocyte zonation markers (Fig. 3d). Only 5.8% of these proteins were expressed equally in all zones (multiple testing-adjusted Shapiro−Wilk test, *P* > 0.05), including electron transfer flavoprotein β (Etfb), the electron acceptor in mitochondrial fatty acid β-oxidation (Extended Data Fig. 6b).

The first principal component along the zonation axis indicated that portal and periportal regions were more similar to one another than central and pericentral zones (Fig. 3b). Indeed, the spatial expression of the top ten significant zonation markers for each portal and central regions followed a hockey-stick curve from portal to central (Fig. 3e), similar to Wnt-controlled transcripts in a scRNAseq dataset[6] and in line with a CV origin of Wnt signaling[14]. In contrast, this pattern was absent for the hits with the highest *P* values (least zonated hits; Extended Data Fig. 6d).

A cross-omics comparison with scRNAseq data[6] confirmed the directionality of the most prominent zonation markers (Pearson's *R* = 0.97, Extended Data Fig. 7a,b), while correlation was low across all proteins and transcripts (Pearson's *R* = 0.12). Notably, a number of proteins were regulated only in the RNA or protein dimension, or

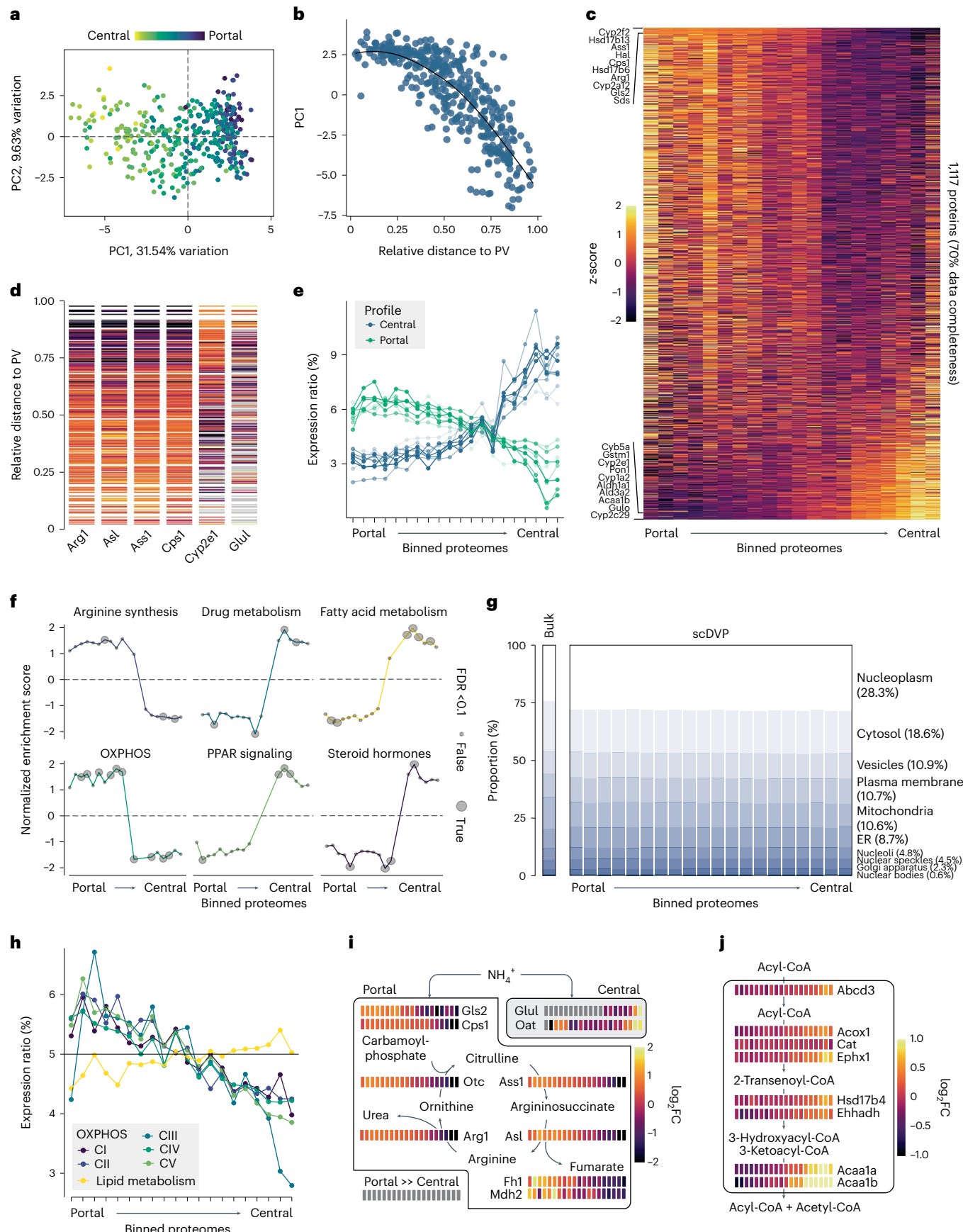

**Fig. 3 | Single-shape proteomes are accurate descriptors of zonated hepatocytes. a**, PCA of all hepatocytes. The color overlay corresponds to the ratio of measured distance PV over CV in the microscopy image. **b**, Measured distance ratio versus PC1. Relative distance of 0 is at the PV and of 1 is at the CV. Black: smoothing curve. **c**, Heat map of protein expression as z-score per protein across all samples. Proteins are ordered according to ANOVA fold change (FC) across 20 spatial equidistant bins, summarizing samples with a similar distance ratio to PV and CV. The ten top and bottom proteins are given. Only proteins that were detected in 70% of all samples are included. **d**, Protein expression as z-score of selected marker proteins, ordered by relative distance to PV and CV. One line is one shape measurement. Gray: protein not detected. **e**, Expression of the top 20 significant proteins in 20 spatial bins, relative to total expression from portal to central. Zonation peak at PV: positive ANOVA fold change ($n = 10$), and vice versa ($n = 10$). **f**, Selected gene sets in individual spatial bins versus all others bins, depicting normalized enrichment score after gene set enrichment analysis. Dot size: significance after multiple testing adjustment. **g**, The proportion of protein signal stratified by subcellular compartment in a bulk mouse liver proteome and the scDVP dataset. Percentages refer to mean across spatial bins in the scDVP data. **h**, Relative expression in 20 bins from PV to CV of proteins constituting mitochondrial OXPHOS components (C) I–V, and mitochondrial lipid metabolism. **i**, Levels of urea cycle and connected enzymes from portal (left) to central (right) bins as $\log_2$FC relative to median expression in the two center bins. Portal box: active in portal regions. Central box: active in central region. **j**, Levels of peroxisomal enzymes related to very-long chain fatty acid degradation, spatially resolved as in **g**. Data from three mice were pooled.

even inversely correlated (Extended Data Fig. 7c), such as dimethylglycine dehydrogenase in the choline catabolic pathway. Similarly, when we compared our data with a FACS-based hepatocyte proteome[8], we found slightly lower correlation of markers and better overlap overall (Pearson's $R$ of 0.16 versus 0.12, Extended Data Fig. 7d–f). Members of glutathione metabolism had similar spatial distribution in both datasets (Extended Data Fig. 7g). This underlines that the scDVP dataset provides orthogonal insight into liver physiology instead of merely complementing existing datasets.

Enrichment of functional protein sets across the spatial bins confirmed that arginine biosynthesis and OXPHOS were highly enriched toward the PV (Fig. 3f). When we added subcellular annotations to our dataset, we found negligible differences to a bulk mouse liver proteome for many compartments including the plasma membrane (summed intensity of 10.6% in the library versus 10.7% in this scDVP data), highlighting that laser microdissection is suitable to excise the entire shape (Fig. 3g and Supplementary Table S6). On a biological level, we found only small changes of summed organellar intensities across spatial bins (Extended Data Fig. 8 and Supplementary Table S6), namely decreasing mitochondrial and endoplasmic reticulum mass and increasing Golgi apparatus and nucleoplasm from PV to CV. When cross-mapping the scDVP data with the mitochondrial protein library Mitocarta 3.0 (ref. 15), the five complexes of OXPHOS decreased collectively by more than 25%, yet mitochondrial proteins related to fatty acid metabolism mildly increased conspatially, suggesting differential regulation within the same cellular compartment (Fig. 3h). Remarkably, these protein sets reach their spatial expression at the midpoint between PV and CV in contrast to the hockey-stick distribution of the top ten differentially expressed proteins (Fig. 3e,h), suggesting that the mitochondrial compartment is not dependent on the Wnt-signaling gradient.

The scDVP data correctly confirmed that proteins participating in ammonia fixation of the urea cycle were highly expressed in portal regions, while those involved in ammonia capture on glutamate were strongly pericentral (Fig. 3i). To our surprise, several other signaling-related pathways were also zonated including peroxisome proliferator-activated receptor (PPAR) signaling (Fig. 3f and Supplementary Table S5). This was corroborated by prominent central expression of enzymes required for peroxisomal degradation of very-long-chain fatty acids, and ω-oxidation of dicarboxylic C12 fatty acids, enriched in, for instance, coconut oil (Fig. 3j). We conclude that the spatial proteome data from single hepatocyte shapes is biologically accurate and informative, and furthermore, contains rich biological information to be mined.

### Spatial context regulates single-cell proteomes

Combining the single-shape proteomes with their inherent spatial information and staining intensities, scDVP revealed clear dependence of fluorescent intensities with the eight proteome classes established above (Fig. 4a,b).

Encouraged by the evident complementarity between extensive proteomics and spatial data, we reasoned that the microscopic image could contain sufficient information to predict the proteome. To this end, we trained a machine-learned (ML) model on 17 features to predict the proteome classes from imaging data. We grouped the training set into five proteome classes by $k$-means clustering (Extended Data Fig. 9a), and used the information in all imaging channels as predictors (Extended Data Fig. 9b). This model reached an average precision of 0.94 (Extended Data Fig. 9c,d), correctly assigning the proteome class of almost all cells. Errors occurred exclusively between spatially neighboring classes (Fig. 4c).

We tested the model performance on a new section (not used in training), from which we measured 60 single-shape proteomes. Visual inspection indicated that the predicted classes were correctly located in proximity to CV or PV, even in the presence of cutting artifacts (Fig. 4d). We used the class probabilities as weights to predict the spatial proteome, which accurately approximated overall protein intensities ($R = 0.78$ between prediction and measurement, Fig. 4e). When predicting the proteome of a larger section for all quantified proteins, the ML model correctly assigned the spatial directionality of zonation markers, as well as their expected extension into the intermediate zone (Fig. 4f). Thus, the model confirms the accuracy of measured single-shape proteomes, and is furthermore a potent predictor of spatial proteomes across any imaged areas.

## Discussion

Here, we present a single-cell spatial map of the murine liver acquired by MS-based proteomics. Our approach successfully combines microscopic imaging data with ultra-high-sensitivity proteomics, building on four major technological advances: (1) artificial intelligence (AI)-assisted segmentation and laser microdissection, (2) multiplex-DIA (mDIA), (3) low-flow gradients and (4) the ultra-high sensitivity of a timsTOF SCP mass spectrometer.

To date, MS-based scProteomics has been exclusively reported for cell suspensions. State-of-the-art workflows currently reach a proteomic depth of up to 2,000 proteins in cultured cells, with about 250 pg of cellular protein mass. This is similar to the protein material in our sliced hepatocytes, taking the section thickness of 10 μm and hepatocyte size of 20–30 μm into account. With our scDVP workflow, we achieved more than 1,700 proteins per single shape (and up to 2,700) despite working from sections that were fixed, stained, imaged and laser dissected. Laser microdissection successfully separated hepatocytes from surrounding material including blood remnants, which holds promise for smaller cell types in more complex tissue environments. The size of our shapes correlated strongly with the number of identified proteins, suggesting that scDVP is currently limited by MS sensitivity and will thus profit from continuous technical developments. We established the scDVP protocol to combine one reference channel with two single shapes (effective two-plex) and used a 40 samples per day chromatography method. This can be

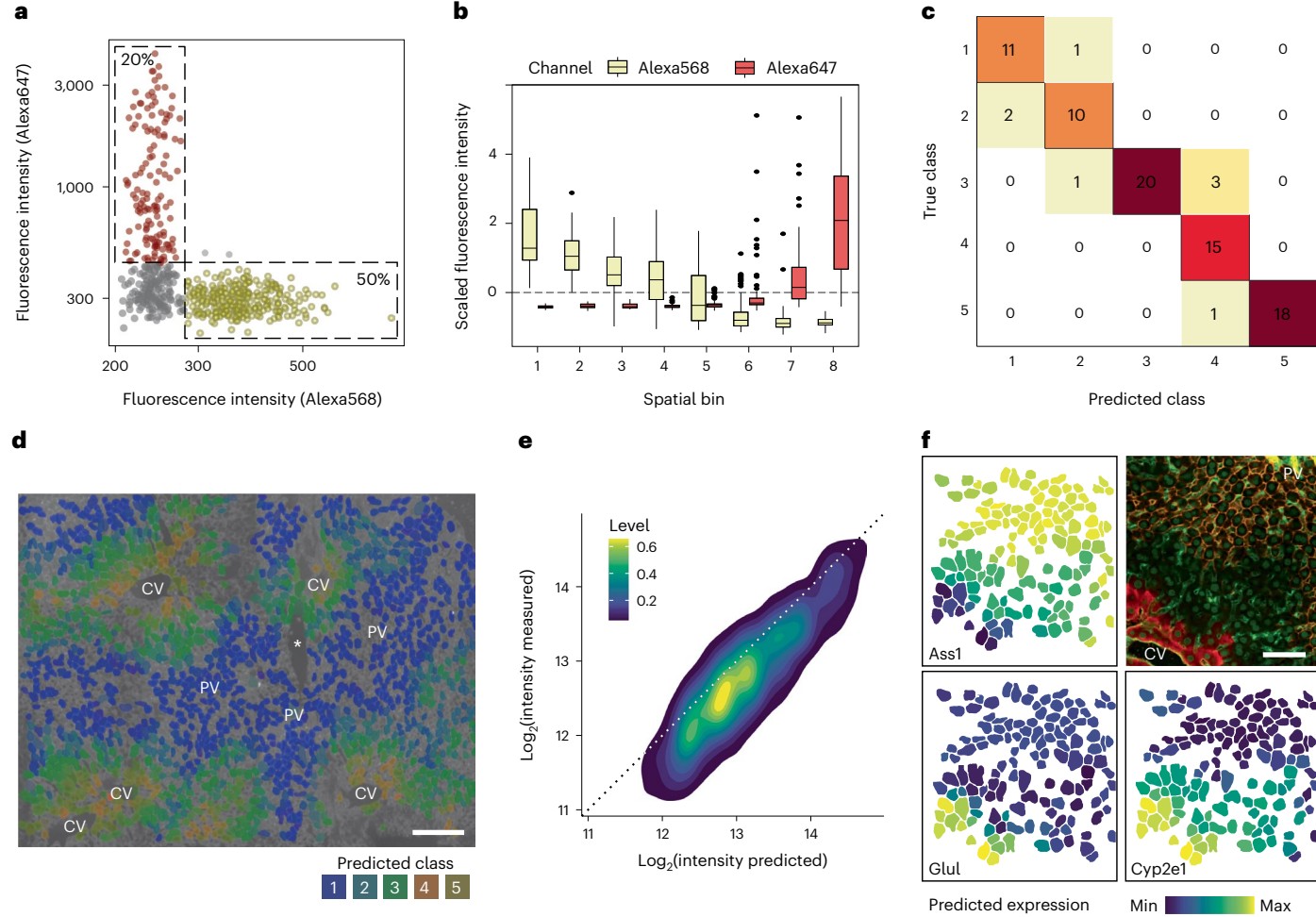

**Fig. 4 | Combining imaging and proteome data for a ML model. a,** Fluorescence intensities of Alexa568 (PV marker E-cadherin) and Alexa647 (CV marker Glul), with percentages in indicated bins. Each dot represents one shape. **b,** Intensities of the spatial markers as in **a** across eight spatial bins. The boxes are first and third quartiles, the thick line is the median, whiskers are ±1.5 interquartile range and outliers are indicated as individual points. **c,** Confusion matrix of a ML model with five classes, informed by microscopy and proteomics data. Colors scale with counts in each box. **d,** Predicted classes of segmented hepatocytes. The hue is the maximum class probability. **e,** Density plot of predicted versus measured intensities of a section excluded from machine learning ($R = 0.78$). **f,** Spatial depiction of one biological replicate with microscopy ground truth on top right, and three predictions. Orange, E-cadherin; red: Glul; green: WGA. *Sectioning artifact. Scale bar, 150 μm.

further scaled to five-plex (effective four-plex) and 80 samples per day, scaling to 320 shapes per day[11]. Given the more stable core proteome compared with single-cell transcriptomes[2] and the resulting lower required sample number, scDVP experiments encompassing a few hundred single shapes could be done in just a few days. Furthermore, because of the very low quantities and absence of proprietary reagents, marginal costs are extremely low.

Our proteomics data from single shapes correctly and accurately recapitulates hepatocyte physiology by direction, extent and spatial organization of zonation. More than half of quantified proteins were significantly different between portal and central zones, in line with scRNAseq data[6,16] and FACS-based proteomics data[8]. The fact that we detected all of the previously used markers of liver zonation[6] suggests that our proteomic depth is sufficient to integrate into other omics datasets. This became further apparent on the level of functional pathways, including signaling and disease pathways. Interestingly, peroxisomal degradation of very-long-chain fatty acids, as well as dicarboxylic C12 fatty acids, was enriched in proximity to the CV. Biochemical evidence by radiolabeling experiments support the notion that nonmitochondrial fatty acid oxidation localizes to pericentral regions[17]. We report an almost linear decrease of mitochondrial mass

and OXPHOS subunits along the zonation axis. This is in line with intra-vital microscopy data showing decreasing mitochondrial membrane potential[18]. A rhythmic expression pattern has been previously shown for a large number of liver transcripts and proteins[16,19]. While we have not covered the temporal aspect here, the scDVP approach could contribute to such studies by adding a spatial dimension.

In the previously described DVP workflow, we used pools of cells combined on the basis of common features, such as the expression intensity of already known markers, or morphology[4]. This approach allows a deep, rapid and robust proteome characterization that accurately represents the underlying biology. By analyzing single cellular shapes without prior assumptions, scDVP now removes the dependency on established markers or features. This makes it a promising approach in heterogeneous tissues with partially or not defined sub-types of cells, such as in many tumor tissues. Moreover, scDVP can be a method of choice to map proteomic disturbances along gradients of, for instance, signaling factors, nutrients or gases, and in physiological settings that may create impediments for other omics methods, for instance, in extracellular fibrotic scars.

Our results demonstrate that the central challenge of scDVP is the sensitivity of the overall workflow. Although we have here reduced the area

required for laser microdissection by 100-fold compared with our initial DVP report[4], we note that one excised hepatocyte shape contains approximately ten times more protein than the smallest cells of interest, such as typical resting lymphocytes[20]. While the required sensitivity is being developed, the original DVP approach using pooled cells of the same type is a powerful tool for this kind of problems. We also note recent success in drastically scaling down DVP for formalin-fixed and paraffin-embedded samples, which are readily available in many clinical settings[21].

There have been advances in the quantification of posttranslational modifications from ultra-low-input material, such as from 1 μg down to the material corresponding to single cell, for instance in the enrichment protocol μPhos[22]. In combination with scDVP, this holds promise for single cells, although the biological information in single-cell phosphoproteomics data would currently be limited to a few high-abundance proteins with high modification stoichiometries. Subtle signaling events, such as the liver-dominant Wnt signaling, will require additional technological developments for in-depth biological description of signaling in single cells by MS.

We have shown that single-cell data can be used to train an accurate ML model that predicts the proteome class from visual information only. Evidence suggests that morphological features such as nuclear vacuolation and texture associate with zonation, and can even serve as a progression and stratification marker of nonalcoholic fatty liver disease[23]. Combining such easily available features and extensive proteomic sampling can clearly lead to higher precision of the predictive models. Transfer learning might then extend the approach to many new areas, as already shown for single-cell transcriptomics data[24]. The modular nature of scDVP, especially the open format from laser microdissection to 384-well plates for sample preparation, makes it widely applicable and also compatible with other spatial omics technologies such as spatial transcriptomics, epigenomics[25] or multiplexed imaging. In conclusion, scDVP is a powerful tool for basic discovery science, working in concert with DVP and other omics methods to enrich spatial workflows.

## Online content

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

## Methods

### Mouse experiments and organ collection

Pathogen-free male and female 10-week-old C57BL/6J-rj mice were purchased from Janvier and maintained at the appropriate biosafety level under constant temperature and humidity conditions with a 12 h light cycle. Animals were allowed food and water ad libitum. All experiments were performed on 12- or 13-week-old wild-type mice. These were killed by cervical dislocation, and the liver was rapidly excised through a ventral opening of the peritoneum. The organ was rinsed in cold phosphate-buffered saline (PBS), and the left lateral lobe was divided into three pieces. For this study, the distal-caudal quarter was embedded in optimal cutting temperature medium (Sakura Finetek) in 15 mm disposable cryomolds (Sakura Finetek) and frozen in isopentane that was precooled to dew point in liquid nitrogen. Fully solidified blocks were transferred to dry ice, and then to a −80 °C freezer until further processing. Animal handling and organ withdrawal were performed in accordance with the governmental and international animal welfare guidelines and ethical oversight by the local government for the administrative region of Upper Bavaria (Germany), registered under ROB-55.2-2532.Vet_02-16-208.

### Immunofluorescence staining

Two micrometer polyethylene naphthalate membrane slides were pretreated by ultraviolet ionization for 1 h at 254 nm. Without delay, slides were consecutively washed for 5 min each in 350 ml acetone and 7 ml VECTABOND reagent to 350 ml with acetone, and then washed in ultrapure water for 30 s before drying in a gentle nitrogen air flow. For sectioning, tissue blocks were transferred to a cryostat (Leica CM3050) at −18 °C chamber and −15 °C object temperature, and left to equilibrate for 30 min. Blocks were then trimmed, and final sections were cut at 10 μm thickness with a disposable high-profile blade (Leica 818). Frozen sections were transferred to pretreated, cold polyethylene naphthalate membrane slides, and melted for less than 5 s on a room temperature surface. The sections were then fixed in prewarmed 4% paraformaldehyde in PBS at 37 °C, then in 95% ethanol at room temperature and finally again in 4% paraformaldehyde in PBS at 37 °C. Slides were rinsed in PBS and left in 5% BSA–PBS blocking solution for 1 h until staining. Sections were stained for 1 h at 37 °C in a humid and dark chamber with 200 μl of a one-step liver painting in 1% BSA: 1:300 phalloidin coupled to Atto-425 (Sigma 66939), 1:200 wheat germ agglutinin (WGA) coupled to Alexa Fluor 488 (Invitrogen, W11261), 1:100 anti-E-cadherin coupled to Alexa Fluor 555 (BD 560064), anti-glutamine synthase (Abcam, ab176562) and 1:500 anti-rabbit nanobody coupled to Alexa Fluor 647 (Chromotek srbAF647-1-100). Slides were washed three times for 2 min in PBS in the dark, and mounted with 21 FL ProLong Diamond mounting medium (Invitrogen, P36961) and a 22 × 22 mm #1.5 coverslip. Slides were stored until imaging in 50 ml tubes with desiccating material at 4 °C.

### High-content imaging

Sections were imaged on an OperaPhenix high-content microscope, controlled with Harmony v4.9 software, at 40× magnification, with binning of two and a per tile overlap of 10%. At excitation wavelengths of 425 nm, 555 nm and 647 nm, an 80% laser intensity were used at an illumination time of 100 ms, while in the 488 nm (CFP) channel, 20% and 20 ms were used. E-cadherin and glutamine synthetase were imaged simultaneously, while phalloidin and WGA were imaged consecutively.

### Image postprocessing

Acquired images were flat-field corrected using the Harmony software. Stitching of image tiles was performed using the ashlar Python API (application programing interface)[26] with a max shift value of 30. Stitched images were exported as .tif files and imported into the Biological Image Analysis Software (BIAS, Single-Cell Technologies Ltd.)[4] with the packaged import tool. In BIAS, large tif images were first retiled

to 1,024 × 1,024 px at an overlap of 5%. Hepatocytes were identified with a deep neural network for histological cytoplasm segmentation on the basis of phalloidin staining at 1.2 input spatial scaling, 40% detection confidence and 30% contour confidence. Only contours between 135 μm² and 1,350 μm² were taken into consideration, and no further exclusion criteria were applied. After removal of duplicates and false identifications by supervised machine learning, contours were exported without additional shape offset together with three calibration points that were chosen at characteristic tissue positions. Contour outlines were simplified by removing 99% of data points. For five-shape proteomes, directly adjacent shapes forming a pentagon-like structure were manually picked. Single shapes were randomly picked and every 15–25th shape was assigned to adjacent wells in a 384-well plate. Arterioles were manually assigned based on WGA signal, ellipticity and proximity to the E-cadherin positive PV.

### Laser microdissection

Contour outlines were imported after reference point alignment, and shapes were cut by laser microdissection with the LMD7 (Leica) in a semi-automated manner at the following settings: power 59, aperture 1, speed 60, middle pulse count 1, final pulse −1, head current 48–52%, pulse frequency 3,282 and offset 100. For the five-shape experiment, the microscope was controlled with LMD v8.2, with which five directly adjacent shapes were sorted into a low-binding 384-well plate (Eppendorf 0030129547) with one empty well between samples. Single shapes were cut and sorted with the software LMD beta 10 after calibration of the gravitational stage shift into 384-well plates into all wells, leaving the outermost rows and columns empty. A 'wind shield' plate was used on top of the sample stage to avoid erroneous sorting. Plates were sealed, centrifuged at 1,000g for 5 min and then frozen at −20 °C until further processing.

### Reference peptide preparation for five-shape and single-shape proteomes

The proximal part of two biologically independent lobes of the same mice as in the scDVP experiments was used to construct a library. The tissue embedded in optimal cutting temperature medium was removed from −80 °C and directly disintegrated in a plastic bag with a manual tissue homogenizer (rubber hammer). Pieces of approximately 1 mm³ were transferred into a low-binding 96-well plate with magnets (Beat-Box Tissue Kit, Preomics), covered with 50 μl of 60 mM triethylammonium bicarbonate buffer with 10% acetonitrile (ACN; lysis buffer), and lysed in a BeatBox (Preomics) at standard settings for 10 min. Samples were then boiled at 96 °C for 20 min, transferred to 1.5 ml low-binding tubes, filled up to 500 μl with lysis buffer and sonicated for five times 30 s on/off cycles. After centrifugation at 2,000g for 1 min, the protein concentration in the supernatant was estimated on a NanoDrop, and LysC and trypsin were added at a protein-to-enzyme ratio of 1:100. After digest for 20 h, samples were acidified to 1% trifluoroacetic acid (TFA), centrifuged at 3,000g for 10 min at room temperature, and dried in a SpeedVac for 30 min. Digest was filled to 1 ml with buffer A (0.1% formic acid (FA)), and desalted on C-18 columns (Waters WAT036820). They were activated and equilibrated with 2 ml of methanol, 2 ml of buffer B (100% ACN, 0.1% FA) and 2 ml of buffer A, before sample loading. Peptides were washed with buffer A two times, eluted in 80% ACN with 0.2% FA and dried down.

### Library fractionation for five-shape proteomes

Peptides were reconstituted in 18 μl buffer A* (0.1% FA, 2% ACN) fractionated on a 30-cm-long 1.9 μm ReproSil C-18 column (PepSep) using a 100 min high-pH gradient. The concentration of buffer B was increased from 3% to 30% in 45 min, to 40% in 12 min, to 60% in 5 min, to 95% in 10 min, kept constant for 10 min, reduced to 5% in 10 min and kept constant for 8 min. The eluted peptides were automatically collected into 48 fractions with a concatenation time of 90 s per fraction. The

fractions were dried in a SpeedVac, reconstituted in 0.1% FA and directly loaded onto Evotips.

## Labeling of single-shape reference proteome
Peptides were reconstituted to 0.125 µg µl$^{-1}$ in 60 mM triethylammonium bicarbonate buffer with 10% ACN, pH 8.5. The peptides were then dimethyl labeled with 0.15% light formaldehyde (CH$_2$O) and 0.023 M light sodium cyanoborohydrate (NaBH$_3$CN) for 1 h at room temperature, quenched with 0.13% ammonia and acidified to 1% TFA. After drying in a SpeedVac, pellets were reconstituted in 100 µl buffer A, and desalted via 5 µg C-18 columns on an AssayMap (Agilent) following the standard protocol. The resulting reference peptides were dried, and reconstituted to 1 ng µl$^{-1}$ in buffer A.

## Peptide preparation of single shapes and dimethyl labeling for multiplexing
Peptides were prepared semi-automated on a Bravo pipetting robot (Agilent), similarly to as described previously[11]. During each incubation step, plates were tightly sealed with two stacked aluminum lids to avoid evaporation (Thermo Fisher Scientific, AB0626). For this, plates were removed from the freezer and centrifuged. The wells were then washed on the robot with 28 µl of 100% ACN and dried in a SpeedVac (Eppendorf) at 45 °C for 20 min. Shapes were then resuspended in 6 µl of 60 mM triethylammonium bicarbonate buffer (pH 8.5, Sigma) with 0.013% DDM (Sigma), and cooked for 30 min at 95 °C in a PCR cycler at a lid temperature of 110 °C. After addition of 1 µl of 80% ACN (final concentration 10%), samples were incubated for another 30 min at 75 °C, cooled briefly, and 1 µl with 4 ng LysC and 6 ng trypsin was added. The samples were digested for 18 h, and then 1 µl of either intermediate (CD$_2$O) or heavy formaldehyde ($^{13}$CD$_2$O) was added to a final concentration of 0.15%. Without delay, either light (NaBH$_3$CN) or heavy (NaBD$_3$CN) sodium cyanoborohydrate were added to 0.023 M to retrieve Δ4 and Δ8 dimethyl-labeled single-shape samples. The sealed plate was then incubated at room temperature for 1 h, and the reaction was quenched to 0.13% ammonia and acidified to 1% TFA.

## Peptide loading onto C-18 tips
C-18 tips (Evotip Pure, EvoSep) were activated for 5 min in 1-propoanl, washed twice with 50 µl of buffer B (99.9% ACN, 0.1% FA), activated for 5 min in 1-propanol, and washed twice with 50 µl buffer A (0.1% formic acid). Single-shape samples were then loaded automatically with the Agilent Bravo robot into 30 µl buffer in the tip that was spun through the C-18 disk for a few seconds only. For loading, 10 µl of 1 ng µl$^{-1}$ reference peptides (Δ0) were pipetted first, followed by Δ4, and Δ8 samples with the same tip. Wells were rinsed with 15 µl buffer A that was also loaded onto the tip. After peptide binding, the disk was further washed once with 50 µl buffer A and then overlayed with 150 µl buffer A. All centrifugation steps were performed at 700g for 1 min, except sample loading for 2 min.

For five-shape proteomes, plates were treated as above, with the exception of lysis in 4.5 µl 60 mM triethylammonium bicarbonate buffer, pH 8.5 without DDM, and consecutive addition of 1 µl LysC and 1.5 µl trypsin to achieve the same digestion volume as above. Five-shape samples were not dimethyl labeled and multiplexed, but acidified directly after digest, and loaded manually onto Evotips following the protocol described above.

## LC–MS/MS analysis of five shapes
Samples were measured with the Evosep One LC system (EvoSep) coupled to a timsTOF SCP mass spectrometer (Bruker Daltonics). The 30 samples per day method was used with the Evosep Performance column 15 cm, 150 µm ID (EV1137 EvoSep) at 40 °C inside a nanoelectrospray ion source (Bruker Daltonics) with a 10 µm emitter (ZDV Sprayer 10, Bruker Daltonics). The mobile phases were 0.1% FA in liquid chromatography (LC)–MS-grade water (buffer A) and 99.9% ACN/0.1% FA (buffer B). We used a dia-PASEF method with 16 dia-PASEF scans separated into four ion mobility windows per scan covering an $m/z$ range from 400 to 1,200 by 25 Th windows and an ion mobility range from 0.6 to 1.6 V s cm$^{-2}$ ('standard scheme'[27]). The mass spectrometer was operated in high sensitivity mode, with an accumulation and ramp time at 100 ms, capillary voltage set to 1,750 V and the collision energy as a linear ramp from 20 eV at $1/K_0 = 0.6$ V s cm$^{-2}$ to 59 eV at $1/K_0 = 1.6$ V s cm$^{-2}$.

## LC–MS/MS analysis of single shapes
Samples were measured with the Evosep One LC system (EvoSep) coupled to a timsTOF SCP mass spectrometer (Bruker Daltonics). The Whisper40 samples per day method was used with the Aurora Elite CSI third generation 15 cm and 75 µm ID (AUR3-15075C18-CS IonOpticks, Australia) at 50 °C inside a nanoelectrospray ion source (Bruker Daltonics). The mobile phases were 0.1% formic acid in LC–MS-grade water (buffer A) and 99.9% ACN/0.1% FA (buffer B). The timsTOF SCP was operated with an optimal dia-PASEF method generated with our Python tool py_diAID[28]. The method contained eight dia-PASEF scans with variable width and two ion mobility windows per dia-PASEF scan, covering an $m/z$ from 300 to 1,200 and an ion mobility range from 0.7 to 1.3 V s cm$^{-2}$, as previously used on the same gradient and similar input material amount[11]. The mass spectrometer was operated in high sensitivity mode, with an accumulation and ramp time at 100 ms, capillary voltage set to 1,400 V and the collision energy as a linear ramp from 20 eV at $1/K_0 = 0.6$ V s cm$^{-2}$ to 59 eV at $1/K_0 = 1.6$ V s cm$^{-2}$.

The labeling efficiency was accessed on the same LC–MS/MS in data-dependent acquisition (dda)-PASEF mode with ten PASEF scans per topN acquisition cycle. Singly charged precursors were excluded by their position in the $m/z$-ion mobility plane using a polygon shape, and precursor signals over an intensity threshold of 1,000 arbitrary units were picked for fragmentation. Precursors were isolated with a 2 Th window below $m/z$ 700 and 3 Th above, as well as actively excluded for 0.4 min when reaching a target intensity of 20,000 arbitrary units. All spectra were acquired within a $m/z$ range of 100–1,700. All other settings were kept as described before.

## Spectral library generation
The spectral library was generated on five dda-PASEF single shots from 50 ng mouse reference peptide, using the same chromatography method as above. Spectra were search with FragPipe v18.0 (ref. [29]) using MSFragger v3.5, Philosopher v4.4.0 and EasyPQP v0.1.32 against a mouse FASTA reference file with 55,319 entries used throughout this study, excluding 50% decoys. Standard settings of the DIA_SpecLib_Quant workflow were used with the following exceptions: N-terminal and lysine mass shift of 28.0313 Da were set as fixed modifications, and methionine oxidation as variable modification. Carbamidomethylation was unselected as samples were not reduced and alkylated. One missed cleavage was accepted. The precursor charge ranged from 2 to 4. The peptide mass range was set to 300–1,800, and peptide length from 7 to 30. For DIA-NN compatibility, the column 'FragmentLossType' was removed in the output library file.

## Spectral search
All 263 files were search together in DIA-NN (version 1.8.1) (ref. [30]) against the above-generated library, using a mass and MS1 mass accuracy of 15.0, scan windows of 9, and activated isotopologues, Match-between-Runs (MBR), heuristic protein inference and no shared spectra, in single-pass mode. Proteins were inferred from genes. Library generation was set as 'IDs, RT & IM profiling', and 'Robust LC (high precision)' as the quantification strategy. Dimethyl labeling at N-termini and lysins was set as fixed modification at 28.0313 Da, and Δ4 or Δ8 were spaced 4.0251 Da or 8.0444 Da from the reference Δ0 ({–fixed-mod Dimethyl, 28.0313, nK} and {–channels Dimethyl, 0, nK, 0:0; Dimethyl, 4, nK, 4.0251:4.0251; Dimethyl, 8, nK, 8.0444:8.0444}). Additional settings were {–original-mods}, {–peak-translation}, {–ms1-isotope-quant}, {–report-lib-info}.

## Data analysis

(1) RefQuant: to determine the quantities of the precursors in the DIA-NN report.tsv file, we utilized the Python-based RefQuant algorithm[11]. In brief, RefQuant determines the ratio between target and reference channels for each individual fragment ion and MS1 peak that is available. This gives a collection of ratios from which RefQuant estimates a likely overall ratio between target and reference. The ratio between target and reference was rescaled by the median reference intensity over all runs for the given precursor, thereby giving a meaningful intensity value for the target channel. The RefQuant quantification matrix was filtered for 'Lib.PG.Q.Value' <0.01, 'Q.value' <0.01 and 'Channel.Q.Value' <0.15 and was then collapsed to protein groups using the MaxLFQ algorithm[31] as implemented in the R package iq (version 1.9.6) (ref. [32]) with median normalization turned off.

(2) Sample filtering and normalization: protein group data were then further analyzed in R v4.2.1 operating in RStudio v2022.07.2. Samples were excluded if the number of detected proteins was below 1.5 or above 3 s.d. from the sample identification median, or within (806, 3,362) identified proteins, resulting into a dropout of 8.9% (41 of 459 samples). Four samples were removed due to their outlier position on the PCA, see Supplementary Table S3. Eight samples were removed due to their cell sizes larger than the BIAS cutoff of 1,350 µm². This resulted in 406 included samples, of which 400 were hepatocytes and 6 endothelial structures for validation. After sample filtering, data was median normalized to a set of proteins that were quantified across all samples (175 proteins quantified in 100% of included samples; Supplementary Table S3), thus correcting for the dependence of protein numbers on shape size. For hepatocyte specific analysis, the arteriole proteomes were removed before normalization.

(3) Figure generation: we chose 20 classes for all comparative spatial analyses as this matches the approximate number of cells from PV to CV, and five classes for machine learning as a compromise between meaningful separation and having enough samples per class. Proteome bins were based on an equidistant split of PC1, distance classes accordingly on a split of PV over CV distance ratios, and applied as indicated. PCA were performed with the PCAtools v2.8.0 package. Limma v3.52.4 was used for statistical testing across proteome bins on a 50%-complete protein data matrix. 'FDR' was applied for multiple testing correction, and an FDR cutoff of 5% was considered significant. Heat mapping was performed with pheatmap 1.0.12, the completeness of the data matrix is indicated in the figure legends. Proteomic gene set enrichment analyses were done with WebGestalt 2019 (ref. [33]) in an R environment using Kyoto Encyclopedia of Genes and Genomes metabolic pathways or Wikipathway as functional library and an FDR threshold for reporting of 1. Significance was defined as FDR <10%, and normalized enrichment scores are reported here. Subcellular localization and mitochondrial functional protein sets were retrieved from mouse Mitocarta 3.0 (ref. [15]). Urea cycle and peroxisomal fatty acid degradation proteins were manually curated. Normality was assessed with Shapiro–Wilk's test, and P values were corrected for multiple testing and expressed as FDR. Spatial data from xml files was plotted with the package sf v1.0-9. For comparisons to scRNAseq data, the dataset of Halpern et al.[6] was used, for which we binned the proteome data into nine equidistant spatial bins as described above. We used the dataset by Ben-Moshe et al.[8] to compare scDVP data with FACS-based proteomics data, binning our samples into eight equidistant spatial bins.

## Image processing

Image data analysis was done in Python (3.8.11). Image shapes were extracted from the stitched tiles using Pillow (9.0.0). For each shape, the bounding box was calculated by taking the floor and ceiling of each shape coordinate and taking the maximum and minimum in x and y. The bounding rectangle was used to crop out the respective region of interest of the image. For image with offset extraction, the center of each bounding rectangle was calculated and rounded to the next integer. An offset of 1,000 was added to each direction to additionally capture the surrounding environment, and the bounding box was highlighted. For composite images, each image was exported per channel with matplotlib (3.5.1), reloaded, merged with NumPy (1.4.2) and saved again. ImageJ was used to manually measure the distance of a shape to its proximal PV and CV.

## Machine learning

For each shape and in all four channels (cyan fluorescent protein, Alexa488, Alexa568 and Alexa647), the mean, median, minimum and maximum intensity of each bounding box were calculated, as well as the shape area. This feature list was saved with pandas (1.22.3). Proteomics data were clustered with a k-means algorithm into five clusters. Next, we used a supervised learning approach to classify the proteomic clusters based on the feature list. The training was performed using the scikit-learn package (1.0.2). Data (n = 408) were randomly split in train and test datasets (split of 0.2). For classification, we used a RandomForestClassifier (n_estimators=200) and achieved a testing accuracy of 0.90. To export probabilities, we used the predict_proba functionality of RandomForest. Diagnostic plots were generated using the Yellowbrick package (1.5). The random state was set to 23 for train/test-split and RandomForestClassifier.

## Reporting summary

Further information on research design is available in the Nature Portfolio Reporting Summary linked to this article.

## Data availability

The mass spectrometry proteomics data have been deposited to the ProteomeXchange Consortium via the PRIDE[34] partner repository with the ID PXD038699. Imaging data has been deposited to BioImages[35] with the accession number S-BIAD596.

## Code availability

The R and Python code used to produce the figures can be downloaded from the Mann lab Github repository via https://github.com/MannLabs/single-cell-DVP.

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

## Acknowledgements

We thank our colleagues at the Department of Proteomics and Signal Transduction at the Max Planck Institute of Biochemistry as well as our colleagues at the Center for Proteome Research in Copenhagen for their input and support. We are particularly grateful for input and help from I. Bludau, A. Brunner and L. Zeitler. We thank Peter H. and Single-Cell Technologies Ltd. for their technical support. F.A.R. is an EMBO postdoctoral fellow (ALTF 399-2021). S.C.M. is a PhD fellow of the Boehringer Ingelheim Fonds. J.G.-S. has received funding from the European Respiratory Society and the European Union's H2020 research and innovation program under the Marie Sklodowska-Curie RESPIRE4 grant agreement no. 847462. This study has been supported by the Horizon-2020 under the MICROB-PREDICT program (M.M., no. 825694) and ISLET (M.M., no. 874839), by the Max Planck Society for Advancement of Science (M.M.), by the Chan Zuckerberg Initiative (M.M., CZF2019-002448) and by grants from the Novo Nordisk Foundation, Denmark (M.M., grant agreements NNF14CC0001 and NNF15CC0001). The funders had no role in study design, data collection and analysis, decision to publish or preparation of the manuscript.

## Author contributions

F.A.R., M.T. and M.M. conceptualized the scDVP workflow. M.T., F.A.R., P.S. and M.W. acquired MS data. F.A.R., M.T.S., L.S., A. Metousis and K.M. performed data analysis. M.T.S. trained the ML model. C.A. developed quantification software. S.C.M. optimized the high-content microscopy pipeline. F.A.R., M.T., L.S., A. Metousis, S.C.M., E.R., T.M.N. and A. Mund developed and optimized the experimental scDVP workflow. J.G.-S., A.S. and H.B.S. provided mouse samples. F.A.R., M.T.S., P.S. and T.M.N. curated data. M.M. and F.A.R. supervised the project. F.A.R. and M.M. wrote the original manuscript draft. All authors read, revised and approved the manuscript.

## Funding

## Competing interests

M.M. is an indirect investor in Evosep. All other authors declare no competing interests.

## Additional information

**Extended data** is available for this paper at https://doi.org/10.1038/s41592-023-02007-6.

**Correspondence and requests for materials** should be addressed to Matthias Mann.

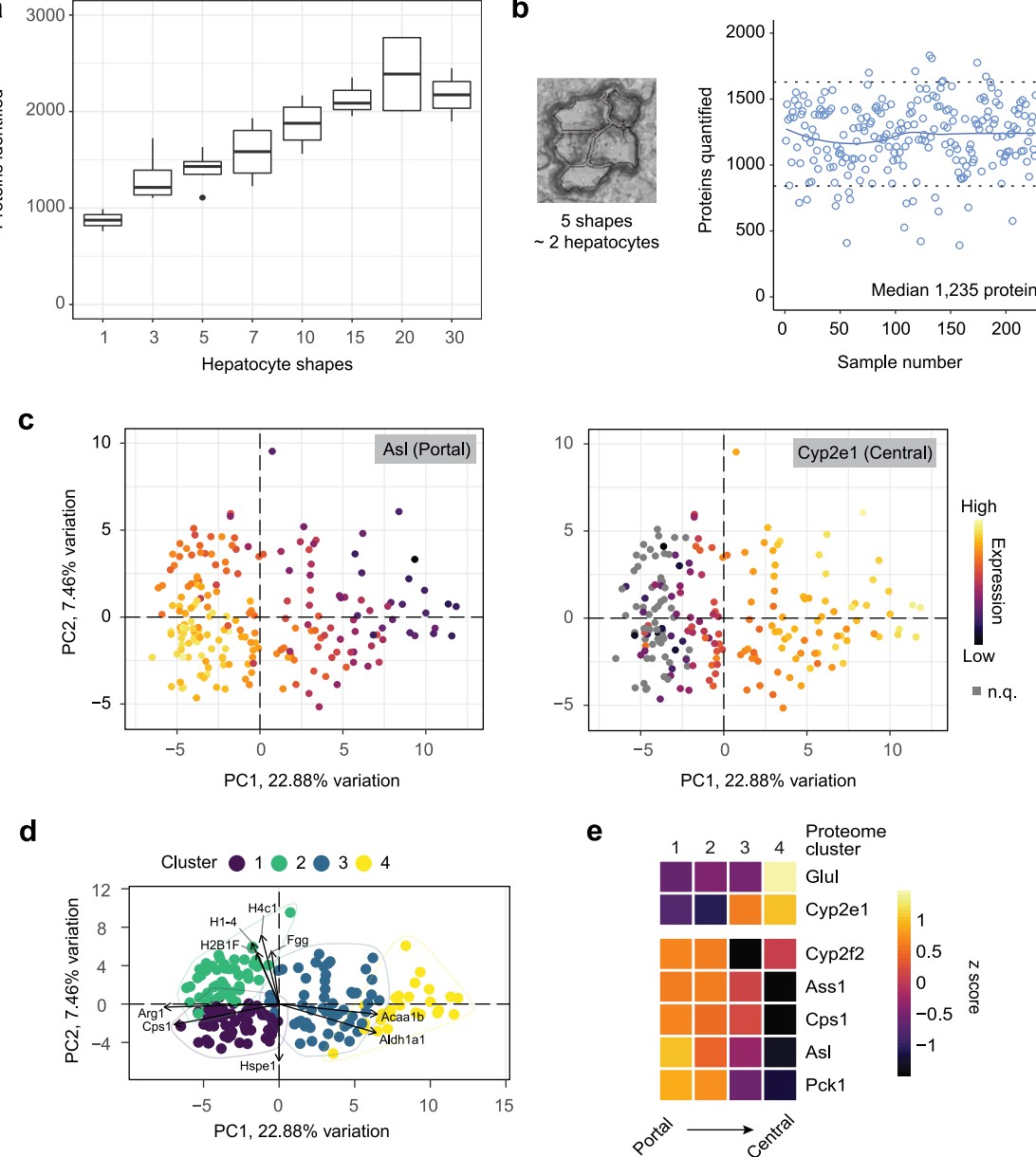

**Extended Data Fig. 1 | Five shape proteomes resolve liver zonation.**
**a**, Titration of number of shapes (10 μm thick) versus proteome depth achieved (n = 3), and measured with the original protocol (single shape, 44 min Evosep gradient, 15 cm column at 500 nL/min, dia-PASEF 27 without optimized windows, library-dependent search in DIA-NN 30). Boxes are first and third quartile, the thick line is median, whiskers are ± 1.5 interquartile range, and outliers are indicated as individual points. **b**, Protein numbers per five shapes across 230 samples. Line is a smoothing curve. **c**, Principal component analyses with a color overlay of two indicated zonation markers; n.q. not quantified. **d**, Unbiased k means clustering of all samples into four bins. Labeled arrows are the top driver proteins of separation. **e**, Marker expression sorted by central (top) or portal (bottom) markers in the indicated k means clusters in d, expressed as z-score of log2 transformed protein abundances, and sorted according to summed zonal probability across all markers.

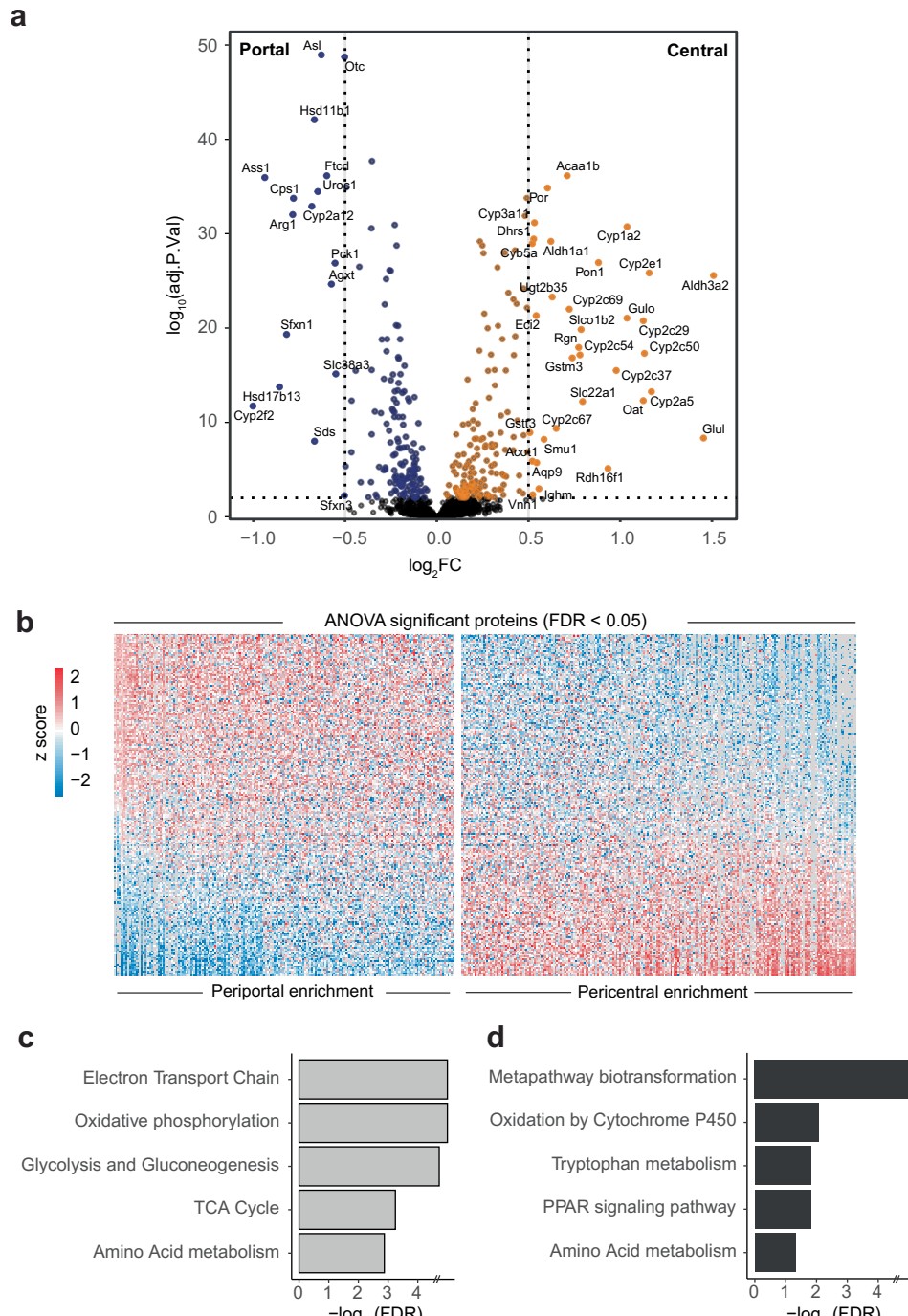

**Extended Data Fig. 2 | Statistical analysis of five-shape proteomes. a**, Volcano plot after an ANOVA over four sorted k means clusters (see Extended Date Fig. 1d). Statistically significant proteins (FDR < 0.05, n = 333 of 1652) with an absolute fold change of more than 0.5 are labeled. Colors indicate upregulation towards portal, or central zones. **b**, Heatmapping of statistically significant proteins in a. The blocks are separate by negative, or positive fold change. Protein expression as z-score of log2 transformed protein abundances. **c,d**, Five top significant terms (FDR < 0.05) after over-representation analysis enriched in peri-portal (**c**) or peri-central regions (**d**). See Supplementary table S2 for further reference.

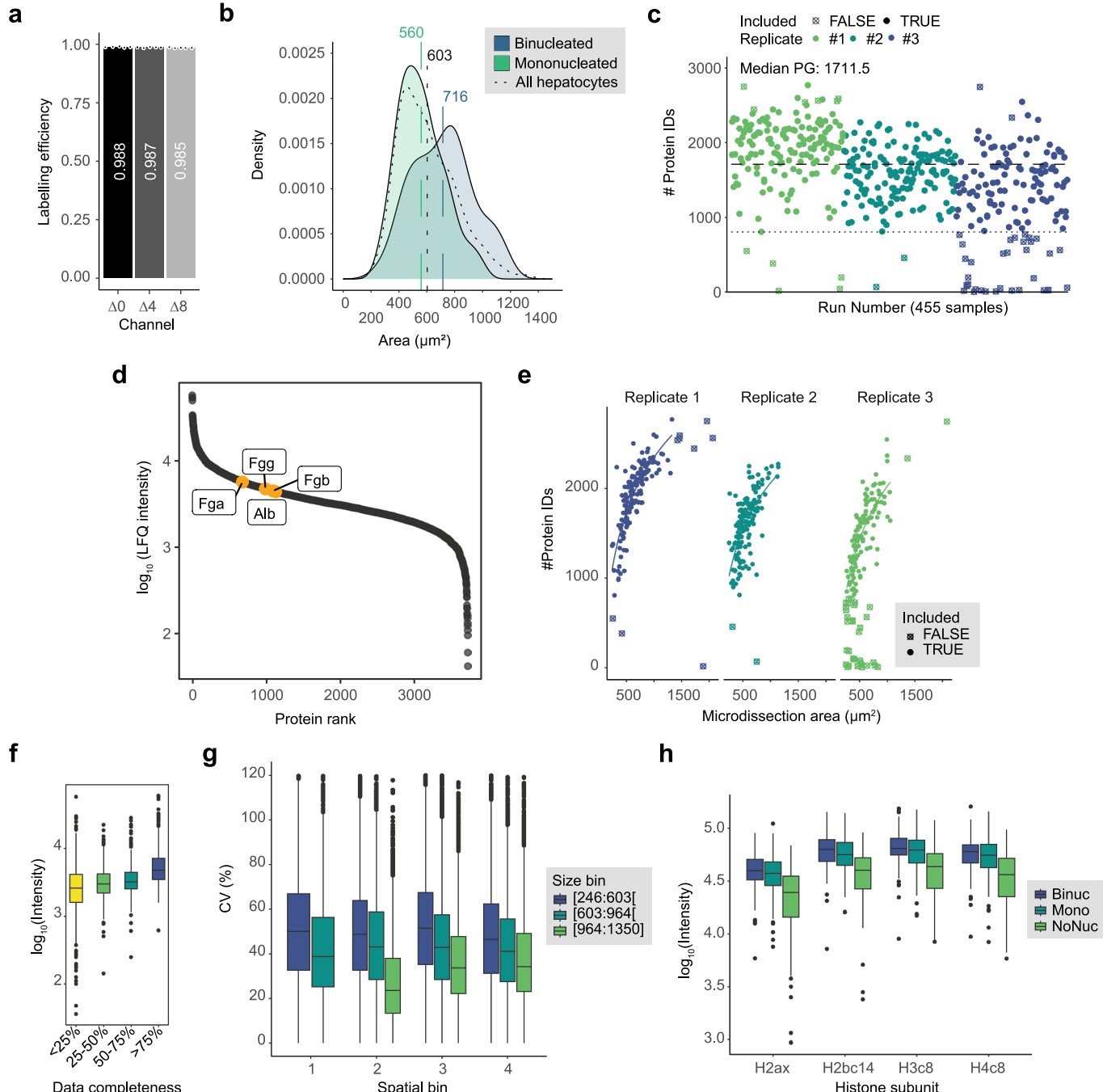

**Extended Data Fig. 3 | Performance overview of single-shape proteomes.**
**a**, Labeling efficiency of 10 ng mouse liver peptide samples. Mean efficiency by intensity is stated in the bar (n = 5, mean and individual measurements). **b**, Density distribution of shape areas across all measured and included hepatocytes, split by visually distinguished mono- (N = 191) and binucleated (N = 99) hepatocytes. Vertical lines and numbers above are mean sizes in the respective group. **c**, Number of proteins per sample (N = 455). The dotted line is the median, the fine pricked line is the sample exclusion cutoff of median minus 1.5 standard deviations. Samples were measured from left to right. Shape type indicates whether the samples was included for the final analysis. **d**, Levels of plasma proteins in the scDVP dataset. Hba, Hbb and Hbd were not detected. **e**, Association between the area of the cut shape, and number of proteins.

Line is a log10 regression curve. Symbols indicate whether sample was included or discarded for analysis, for exclusion criteria see Methods section. **f**, Percentage of proteins quantified, binned into four groups, versus log10 transformed median intensities in the respective bin. Data completeness is defined as percentage of samples across all samples in which a particular protein was quantified in. **g**, Coefficient of variation (CV) in bins of similarly sized shapes (color coded), and spatial bins with similar distance ratio to portal and central vein, that is similar zonation profile. **h**, Levels of four histone proteins shown in Fig. 2b by number of nuclei in the isolated shapes. Binuc: binucleated (N = 99); Mono: mononucleated (N = 191); NoNuc: no nucleus (N = 101). Boxes are first and third quartile, the thick line is median, whiskers are ± 1.5 interquartile range, and outliers are indicated as individual points.

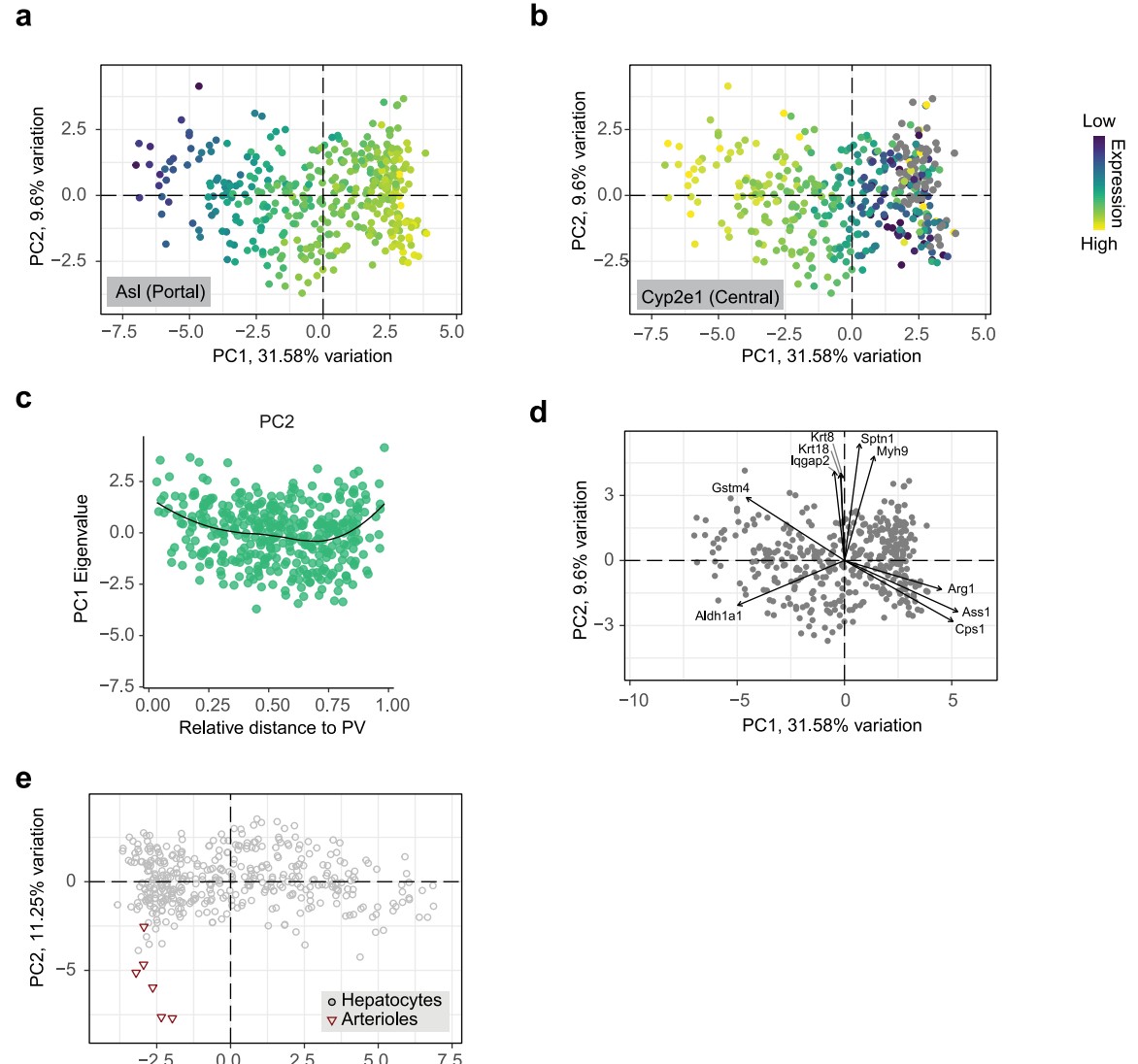

**Extended Data Fig. 4 | Dimensionality reduction of single shape data.**
**a**, Color overlay is expression level of the portal marker Asl, or **b**, the central marker Cyp2e1. **c**, PC2 versus measured distance ratio portal over central vein for all shapes. **d**, Top 10-leading edges as Eigenvectors (arrows) with proteins. **e**, Arterioles were cut as quality controls (see Methods section), and separate from hepatocytes on PC2 (n = 6).

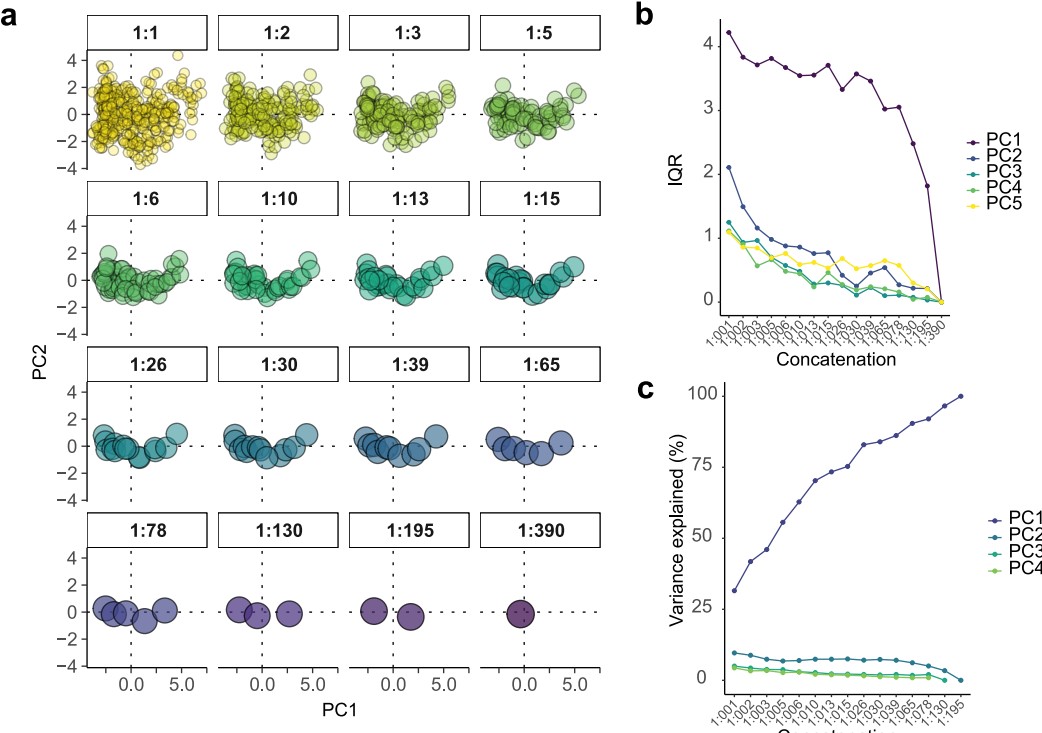

**Extended Data Fig. 5 | Information aggregation from single shapes.**
**a**, Principal component analysis after averaging of close-by cells, as measured by relative position along the portal to central vein zonation axis. Ratios over every sub-plot indicate concatenation ratio (1:x averages x cells). **b**, Interquartile range (IQR) of principal components 1 – 5 at given concatenation ratio. **c**, Variance explained by the indicated principal component at given concatenation ratio.

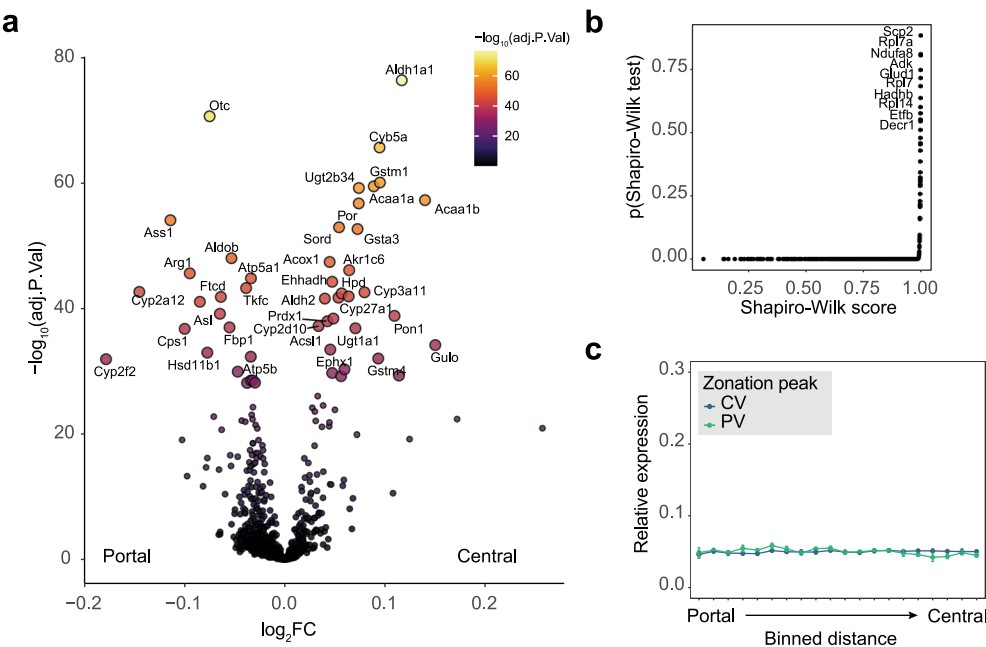

**Extended Data Fig. 6 | Functional analysis of single shape data. a**, Volcano plot after ANOVA across 20 spatially guided bins. Color overlay specifies adjusted p value, the top 40 significant proteins are labeled. **b**, Score and multiple testing-adjusted p value of a Shapiro-Wilk normality test. Lowest proteins are labeled. **c**, Relative expression normalized to 1 for each contributing protein (n = 10) of the least significant Shapiro-Wilk hits in b, from portal to central distance-guided bins.

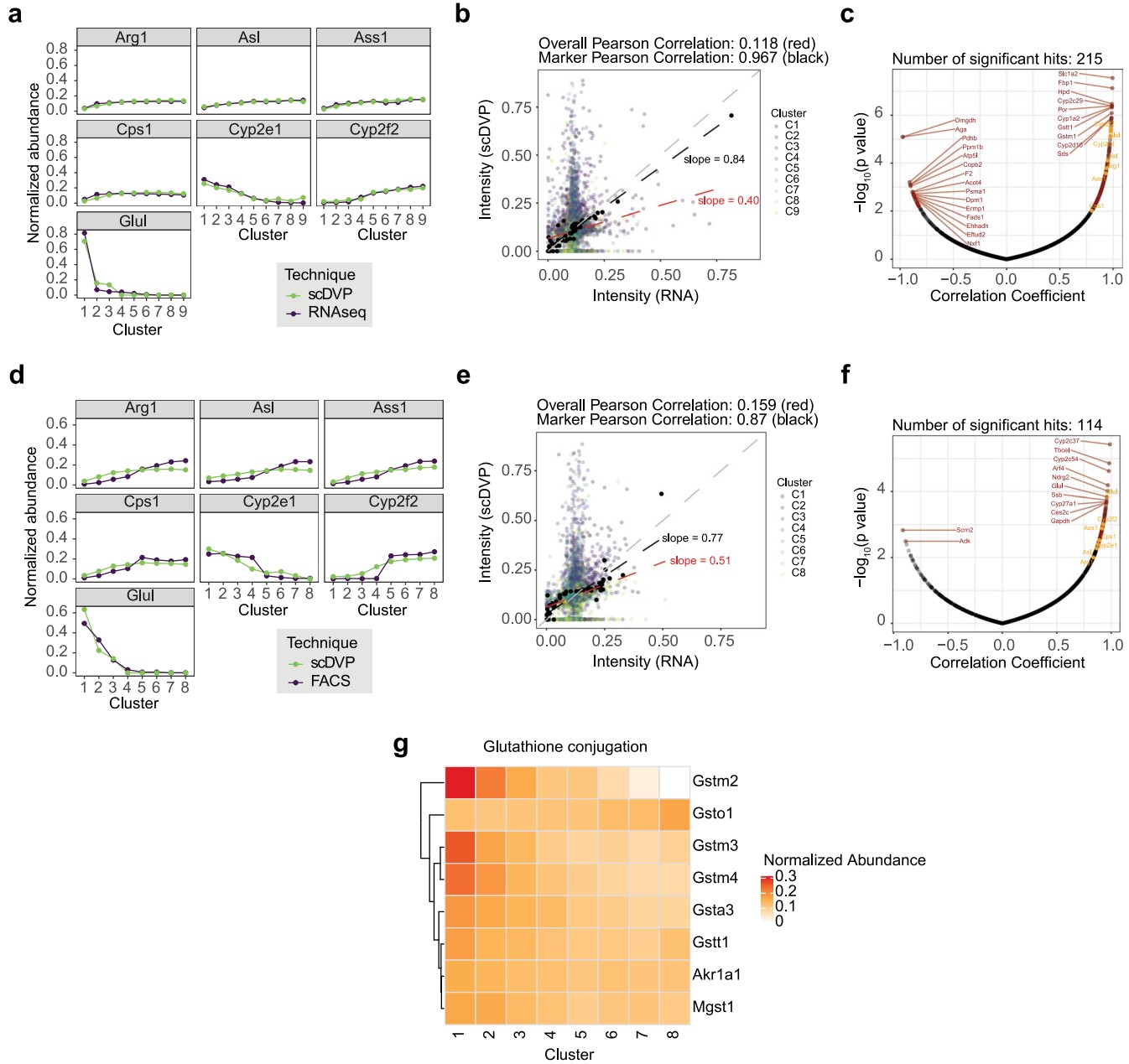

**Extended Data Fig. 7 | Comparison of scDVP to existing scRNAseq data (a-c) and FACS-based proteomics data (d-g). a**, Abundance normalized to 1 across 9 bins in Halpern et al. 6 (marker expression-guided bins), and this scDVP data (spatial bins). **b**, Intensity correlation of all hits (opaque dots, color according to cluster) and markers (black dots). Linear regression as dashed line, with Pearson correlation coefficient given over the figure. Grey line is the 45 degree line. **c**, Correlation coefficient for targets across all bins, with multiple testing adjusted p value. Top hits on either side are labeled in dark red, and marker proteins in orange. **d**, Abundance normalized to 1 across 8 bins in Ben-Moshe et al. 8 (marker expression-guided bins), and this scDVP data (spatial bins). **e**, Intensity correlation of all hits (opaque dots, color according to cluster) and markers (black dots). Linear regression as dashed line, with Pearson correlation coefficient given over the figure. Grey line is the 45 degree line. **f**, Correlation coefficient for targets across all bins, with multiple testing adjusted p value. Top hits on either side are labeled in dark red, and marker proteins in orange. **g**, A significant hit after gene set enrichment analysis on Pearson correlation coefficients, with normalized abundance of protein levels as heatmap colors.

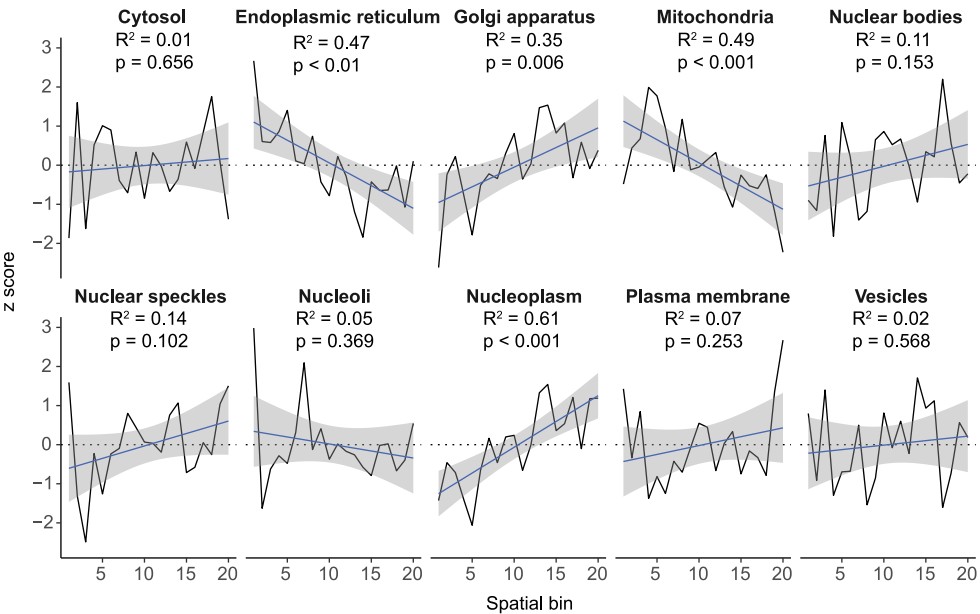

**Extended Data Fig. 8 | Changes in subcellular compartment composition across space.** Spatial bins are mean single shape data in 20 equidistant bins from portal to central vein. Ordinate values are z-transformed proportions of summed signal intensities per compartment. Pearson's R was calculated on z scores from a linear model. Blue line is the linear regression line with the 95% confidence interval in grey.

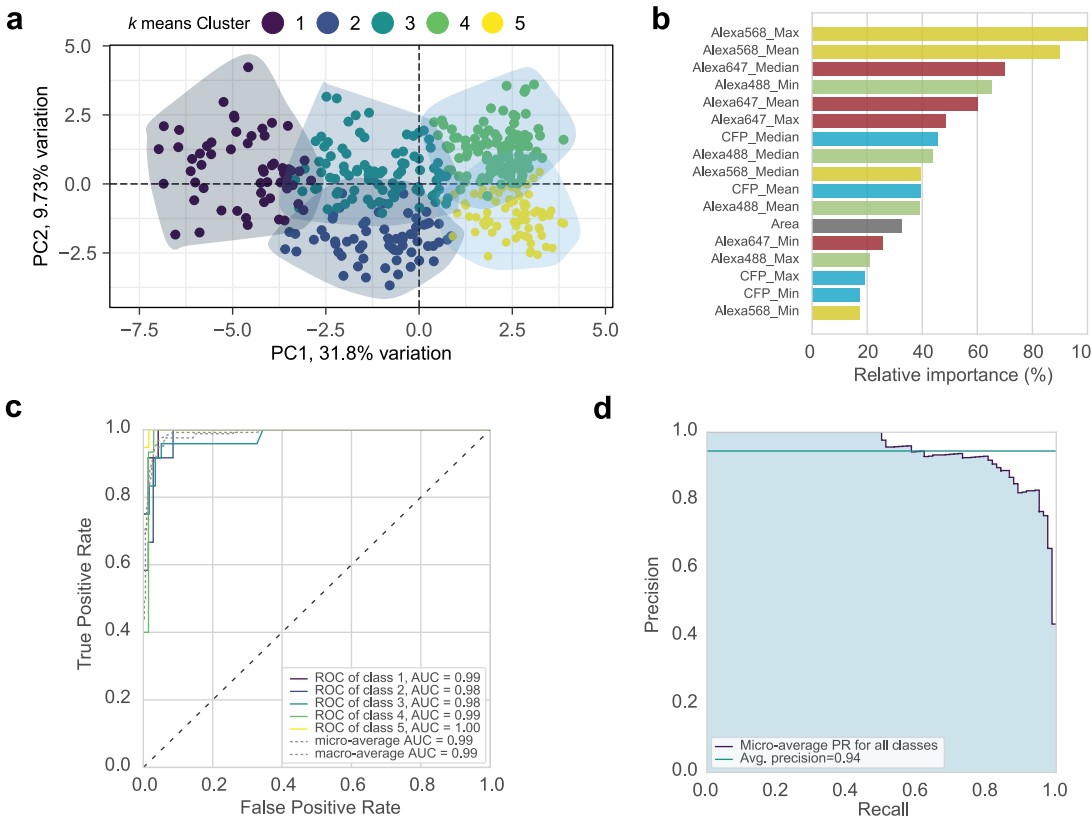

**Extended Data Fig. 9 | Machine learning (ML) accurately predicts proteome class. a**, $k$ means clustering, dividing all samples into five classes that inform the ML. **b**, Feature importance of the ML model, relative to the highest contributor. **c**, Receiver-Operating-Characteristics for each class. The individual Area Under the Curve (AUC) is given in the graph. **d**, Precision-recall-curve for the five classes.

# Reporting Summary

## Statistics

For all statistical analyses, confirm that the following items are present in the figure legend, table legend, main text, or Methods section.

| n/a | Confirmed | |
|---|---|---|
| ☐ | ☒ | The exact sample size (*n*) for each experimental group/condition, given as a discrete number and unit of measurement |
| ☐ | ☒ | A statement on whether measurements were taken from distinct samples or whether the same sample was measured repeatedly |
| ☐ | ☒ | The statistical test(s) used AND whether they are one- or two-sided<br>*Only common tests should be described solely by name; describe more complex techniques in the Methods section.* |
| ☐ | ☒ | A description of all covariates tested |
| ☐ | ☒ | A description of any assumptions or corrections, such as tests of normality and adjustment for multiple comparisons |
| ☐ | ☒ | A full description of the statistical parameters including central tendency (e.g. means) or other basic estimates (e.g. regression coefficient) AND variation (e.g. standard deviation) or associated estimates of uncertainty (e.g. confidence intervals) |
| ☐ | ☒ | For null hypothesis testing, the test statistic (e.g. *F*, *t*, *r*) with confidence intervals, effect sizes, degrees of freedom and *P* value noted<br>*Give P values as exact values whenever suitable.* |
| ☒ | ☐ | For Bayesian analysis, information on the choice of priors and Markov chain Monte Carlo settings |
| ☒ | ☐ | For hierarchical and complex designs, identification of the appropriate level for tests and full reporting of outcomes |
| ☐ | ☒ | Estimates of effect sizes (e.g. Cohen's *d*, Pearson's *r*), indicating how they were calculated |

*Our web collection on statistics for biologists contains articles on many of the points above.*

## Software and code

Policy information about availability of computer code

| | |
|---|---|
| Data collection | Commercially available: Perkin Elmer Harmony 4.9, Biological Image Analysis Software BIAS v2022-02-02 (Single-Cell Technologies Ltd., ref 4 and acknowledgements), Leica LMD 8.2, Leica LMD beta 10, Bruker HyStar 6.0, timsControl 3.0.20, Evosep One RCNet Driver 2.2.74.0. Open source: ashlar python API (ref 26) |
| Data analysis | FragPipe 18.0 (ref 29), MSFragger 3.5 Philosopher 4.4.0, EasyPQP 0.1.32, DIA-NN 1.8.1 (ref 30), MaxQuant 2.1.3.0, R 4.2.1 [packages: WebGestaltR 0.4.4, ggridges 0.5.4, RColorBrewer 1.1-3, sf 1.0-9, XML 3.99-0.12, rstatix 0.7.0, MASS 7.3-57, devtools 2.4.5, org.Mm.eg.db 3.15.0, AnnotationDbi 1.58.0, IRanges 2.30.1, pheatmap 1.0.12, limma 3.52.4, viridis 0.6.2, viridisLite 0.4.1, ggrepel 0.9.1, forcats 0.5.2, stringr 1.4.1, dplyr 1.0.10, purrr 0.3.5, readr 2.1.3, tidyr 1.2.1, tibble 3.1.8, ggplot2 3.3.6, tidyverse_1.3.2], RStudio 2022.07.2, Python 3.8.11 [packages: Pillow 9.0.0, Numpy 1.4.2, pandas 1.22.3, scikit-learn 1.0.2, Yellowbrick 1.5]. Custom open source: py_diAID (ref 28), RefQuant (ref 11). |

For manuscripts utilizing custom algorithms or software that are central to the research but not yet described in published literature, software must be made available to editors and reviewers. We strongly encourage code deposition in a community repository (e.g. GitHub). See the Nature Portfolio guidelines for submitting code & software for further information.

## Data

Policy information about availability of data

All manuscripts must include a data availability statement. This statement should provide the following information, where applicable:

- Accession codes, unique identifiers, or web links for publicly available datasets
- A description of any restrictions on data availability
- For clinical datasets or third party data, please ensure that the statement adheres to our policy

> The mass spectrometry proteomics data have been deposited to the ProteomeXchange Consortium via the PRIDE 34 partner repository with the ID PXD038699. Imaging data has been deposited to BioImages with the accession number S-BIAD596.

## Human research participants

Policy information about studies involving human research participants and Sex and Gender in Research.

| | |
|---|---|
| Reporting on sex and gender | NA |
| Population characteristics | NA |
| Recruitment | NA |
| Ethics oversight | NA |

Note that full information on the approval of the study protocol must also be provided in the manuscript.

# Field-specific reporting

Please select the one below that is the best fit for your research. If you are not sure, read the appropriate sections before making your selection.

☒ Life sciences  ☐ Behavioural & social sciences  ☐ Ecological, evolutionary & environmental sciences

For a reference copy of the document with all sections, see nature.com/documents/nr-reporting-summary-flat.pdf

# Life sciences study design

All studies must disclose on these points even when the disclosure is negative.

| | |
|---|---|
| Sample size | Five-shape proteomes: n = 5 mice, 230 samples. Single-shape proteomes: n = 3 mice, 459 samples; one validation mouse with 54 additional samples. No sample size calculations were performed, but estimated to full 10 days of full time MS measurements ( > samples). |
| Data exclusions | Single-shape samples were excluded if the number of proteins was below 806 or above 3362 proteins (median number of proteins - 1.5 SD or + 3 SD; discarded 42, kept 418). Four samples were removed due to their outlier position in the principal component analysis (discarded 4: m3B_14_target8, m3B_40_target4, m4A_58_target8, m4A_67_target4; kept 414). Eight samples were removed due to their cell sizes larger than 1350 μm2 (kept 406). |
| Replication | Each single-cell slice per mouse was treated as an individual replicate, thus amounting to a total of 400 hepatocytes and 6 arterioles. Batch correction was applied in an unsupervised way via RefQuant in relation to the reference proteome channel. Fifty-four additional samples from a separate mouse were used as a validation of the data. |
| Randomization | Allocation of samples was performed randomly and in an unsupervised way. Of all segmented hepatocytes, every 15th, 20th or 25th (depending on the total number of hepatocytes per section to reach wide coverage across one section) was cut and measured. The samples of one mouse were prepared and measured in one batch (in the order m5C, m4A, m3B, m1A) as samples should be as fresh as possible when analyzed by LC-MS/MS. |
| Blinding | Investigators were not blinded due to the study design, that is experimentally verifying ground truth to confirm robustness of the method. Several steps were unsupervised to eliminate observer bias: mice were sacrificed in a random order, selection of hepatocytes was random (every 15th, 20th or 25th of all segmented hepatocytes depending on total number of cells per section), samples were excluded according to a calculated cutoff (see data exclusions). -- Exclusion based on PCA position (n = 4 of 414 samples) was not biased. |

# Reporting for specific materials, systems and methods

We require information from authors about some types of materials, experimental systems and methods used in many studies. Here, indicate whether each material, system or method listed is relevant to your study. If you are not sure if a list item applies to your research, read the appropriate section before selecting a response.

## Materials & experimental systems

| n/a | Involved in the study |
|-----|----------------------|
| ☐ | ☒ Antibodies |
| ☒ | ☐ Eukaryotic cell lines |
| ☒ | ☐ Palaeontology and archaeology |
| ☐ | ☒ Animals and other organisms |
| ☒ | ☐ Clinical data |
| ☒ | ☐ Dual use research of concern |

## Methods

| n/a | Involved in the study |
|-----|----------------------|
| ☒ | ☐ ChIP-seq |
| ☒ | ☐ Flow cytometry |
| ☒ | ☐ MRI-based neuroimaging |

## Antibodies

| Antibodies used | Anti-e cadherin coupled to Alexa Fluor 555 (BD 560064) used at 1:100<br>Anti-glutamine synthase (rabbit, Abcam ab176562) used at 1:200<br>Anti-rabbit nanobody coupled to Alexa Fluor 647 (Chromotek srbAF647-1-100) used at 1:500 |
|-----------------|--------|
| Validation | BD 560064: verified by manufacturer with Western blotting using purified mouse E-Cadherin and overpression of human E-Cadherin in 293F cells<br>ab176562: validated by manufacturer with Western blotting, and a HAP1 KO cell line |

## Animals and other research organisms

Policy information about studies involving animals; ARRIVE guidelines recommended for reporting animal research, and Sex and Gender in Research

| Laboratory animals | Pathogen-free male and female 10-week-old C57BL/6J-rj mice were purchased from Janvier (France) and maintained at the appropriate biosafety level under constant temperature and humidity conditions with a 12h light cycle. Animals were allowed food and water ad libitum. All experiments were performed on 12- or 13-week-old wild-type mice. |
|--------------------|--------|
| Wild animals | No wild animals were used in this study. |
| Reporting on sex | Only male mice were used due to the study design, i.e. validation of a method. |
| Field-collected samples | No field-collected samples were used in this study. |
| Ethics oversight | Animal handling and organ withdrawal were performed in accordance with the governmental and international animal welfare guidelines and ethical oversight by the local government for the administrative region of Upper Bavaria (Germany), registered under ROB-55.2-2532.Vet_02-16-208. |

Note that full information on the approval of the study protocol must also be provided in the manuscript.

