## [Peer Review File · Nature Methods]

Peer Review Information

Manuscript Title: Spatial single-cell mass spectrometry defines zonation of the hepatocyte proteome

Corresponding author name(s): Matthias Mann

Editorial Notes: n/a

Reviewer Comments & Decisions:

Decision Letter, initial version:

Dear Matthias,

Your Article, "Spatial single-cell mass spectrometry defines zonation of the hepatocyte proteome", has now been seen by 3 reviewers. As you will see from their comments below, although the reviewers find your work of considerable potential interest, they have raised a number of concerns. We are interested in the possibility of publishing your paper in Nature Methods, but would like to consider your response to these concerns before we reach a final decision on publication.

We therefore invite you to revise your manuscript to address these concerns. We recommend you provide all technical details pertaining to the method as mentioned by the reviewers, discuss limitations, and tone down some of the language as pointed out by reviewer #3. We also think an additional demonstration on a more challenging sample would really strengthen the paper.

* include a point-by-point response to the reviewers and to any editorial suggestions

* please underline/highlight any additions to the text or areas with other significant changes to facilitate review of the revised manuscript

* address the points listed described below to conform to our open science requirements

* ensure it complies with our general format requirements as set out in our guide to authors at www.nature.com/naturemethods

* resubmit all the necessary files electronically by using the link below to access your home page

[URL] This URL links to your confidential home page and associated information about manuscripts you may have submitted, or that you are reviewing for us. If you wish to forward this email to co-authors, please delete the link to your homepage.

If you cannot send it within this time, please let us know. In this event, we will still be happy to reconsider your paper at a later date so long as nothing similar has been accepted for publication at Nature Methods or published elsewhere.

OPEN SCIENCE REQUIREMENTS

REPORTING SUMMARY AND EDITORIAL POLICY CHECKLISTS

IMAGE INTEGRITY

DATA AVAILABILITY

All novel DNA and RNA sequencing data, protein sequences, genetic polymorphisms, linked genotype and phenotype data, gene expression data, macromolecular structures, and proteomics data must be deposited in a publicly accessible database, and accession codes and associated hyperlinks must be provided in the "Data Availability" section.

To further increase transparency, we encourage you to provide, in tabular form, the data underlying the graphical representations used in your figures. This is in addition to our data-deposition policy for specific types of experiments and large datasets. For readers, the source data will be made accessible directly from the figure legend. Spreadsheets can be submitted in .xls, .xlsx or .csv formats. Only one (1)

file per figure is permitted: thus if there is a multi-paneled figure the source data for each panel should be clearly labeled in the csv/Excel file; alternately the data for a figure can be included in multiple, clearly labeled sheets in an Excel file. File sizes of up to 30 MB are permitted. When submitting source data files with your manuscript please select the Source Data file type and use the Title field in the File Description tab to indicate which figure the source data pertains to.

Please include a “Data availability” subsection in the Online Methods. This section should inform readers about the availability of the data used to support the conclusions of your study, including accession codes to public repositories, references to source data that may be published alongside the paper, unique identifiers such as URLs to data repository entries, or data set DOIs, and any other statement about data availability. At a minimum, you should include the following statement: “The data that support the findings of this study are available from the corresponding author upon request”, describing which data is available upon request and mentioning any restrictions on availability. If DOIs are provided, please include these in the Reference list (authors, title, publisher (repository name), identifier, year). For more guidance on how to write this section please see: <http://www.nature.com/authors/policies/data/data-availability-statements-data-citations.pdf>

CODE AVAILABILITY

Please include a “Code Availability” subsection in the Online Methods which details how your custom code is made available. Only in rare cases (where code is not central to the main conclusions of the paper) is the statement “available upon request” allowed (and reasons should be specified).

For more information on our code sharing policy and requirements, please see: <https://www.nature.com/nature-research/editorial-policies/reporting-standards#availability-of-computer-code>

MATERIALS AVAILABILITY

SUPPLEMENTARY PROTOCOL

To help facilitate reproducibility and uptake of your method, we ask you to prepare a step-by-step Supplementary Protocol for the method described in this paper. We [encourage authors to share their step-by-step experimental protocols](https://www.nature.com/nature-research/editorial-policies/reporting-standards#protocols) on a protocol sharing platform of their choice and report the protocol DOI in the reference list. Nature Portfolio's Protocol Exchange is a free-to-use and open resource for protocols; protocols deposited in Protocol Exchange are citable and can be linked from the published article. More details can found at www.nature.com/protocolexchange/about.

ORCID

Sincerely,
Arunima

Arunima Singh, Ph.D.
Senior Editor
Nature Methods

Reviewers' Comments:

Reviewer #1:

Remarks to the Author:

Review comments

In the current study, Rosenberger et al. used an approach of combined imaging, laser capture microdissection and MS to characterize the spatial proteome of small cellular fraction of murine hepatocytes. The authors modified their previously published protocol on single cells proteomics applied on suspended cells to allow single-cell proteomics evaluation with spatial context. The authors combined several staining markers to identify the liver lobule architecture. They first assessed what is the minimum area to collect to achieve significant and reproducible proteome measurement. Next, they collected 455 single shape samples and binned them to 8 zones, and found that the proteome measurements of these shapes recapitulate liver lobule biology of previously zoned genes. Finally, the authors developed an AI tool to predict single shape proteome based on microscopic images. This is the first demonstration of single cell proteomics in an intact mammalian tissue, an impressive achievement that warrants publication in Nature Methods.

Several points should be addressed in the revision:

1. In general the data supplied do not enable easily reconstructing the analysis and applying new analyses. The authors should provide: a) Their reconstructed pc coordinates for each cell. b) The measured distance to the portal veins and central veins for each cell (shown in figure 3b). c) the nuclear size, as estimated from the catapulted cellular image. This will enable easy analysis of protein zonation, as well as protein changes between hepatocytes of different ploidy classes.
2. Related to the above point, hepatocytes at the analyzed ages are either diploid, tetraploid or octoploid, while the authors analyzes thin tissue sections, nuclear diameter can still be used to at least estimated ploidy, are there differences between the proteomes of hepatocytes of different ploidies that reside within the same zone? Hepatocyte ploidy impacts gene expression in an only marginal manner (<https://pubmed.ncbi.nlm.nih.gov/34253736/>), it would be interesting to explore the impact on the hepatocyte proteome, given the new tool presented.
3. Figure 1a middle panel suggests that delta4 and delta8 pertain to mono or bi-nucleated hepatocytes, is this the case? Please explain better.
4. Zonation reconstruction – why do the authors use the PC1 rather than the absolute distances to the portal/central veins? It seems these were recorded for each cell.
5. The authors compared their data to RNA zonation, please compare also to protein zonation obtained via spatial sorting in <https://www.nature.com/articles/s42255-019-0109-9>. It seems like the dynamic range in the current single cell dataset is lower than the spatial sorting study. Is there some background intensity that should be removed from the proteome? In addition, while zonation trends between the

two studies nicely match, absolute levels of proteins do not, can the authors provide some methodological explanation to such discordances?

6. In general, aspects of data analysis description in the paper is missing, would be important to elaborate on the pre-processing.

7. The authors' data nicely show mid-lobule zonation of genes such as *Hamp2* and *Cyp8b1*, previously shown to have non-monotonic zonation patterns (<https://www.nature.com/articles/nature21065>). I would mention this in the text.

8. "Only contours between 30 μm^2 and 300 μm^2 " – in 13w old mice is there such a diversity in cells size? How did you make sure you sample only hepatocytes?

Reviewer #2:

Remarks to the Author:

Rosenberger et al., provides an update of the Deep Visual Proteomics method that was previously developed by the Mann lab. This technical update offers increased sensitivity, which allows the authors to achieve single cell resolution from . Main elements of this critical improvement were: use of surfactant n-Dodecyl- β -D-maltoside (DDM) to maximize peptide recovery, lowering the chromatographic flow rate for increased ionization efficiency, achieving higher chromatographic resolution with zero dead volume columns, and addition of a labeled reference channel for multiplexed data-independent acquisition (DIA) that decouples identification and quantification.

The improved method is verified by a demonstration that captures the spatial zonation of the hepatocyte proteome in the mouse liver tissue along the portal-central vein axis, including a comparison to previously published single cell RNA sequencing data. Using the acquired spatial proteomics data, the authors also train an AI model to predict proteome classes from imaged parameters.

The current work can be seen as an exciting extension of the Mund et al., 2022, Nature Biotechnology, which was a big leap towards untargeted spatial proteomics. It is mainly driven by optimization of the sample preparation workflow to increase the sensitivity and hence allow proteomic analysis from lower inputs corresponding roughly to single hepatocytes. I think the promise of the method for realizing untargeted spatial proteomics from fixed tissue samples is very high and I am happy to see the DVP method reaching a higher performance. Overall, I think the scDVP is definitely the cutting-edge of this class of technologies and this first attempt to utilize AI for predicting proteome classes from images is also quite interesting.

Having said that, major general weaknesses I see with the manuscript are:

Although it is a technical update over the first DVP implementation, some of the relevant technical details about the technical improvements, critical considerations for assessment of the results and potential limitations of the method are glossed over.

The implementation on the liver zonation, although looking very sound, may not be the best benchmarking/demonstration case to highlight the capabilities of the method as proteome profiles of hepatocytes (being rather large cells) with clear metabolic differences along the chosen spatial axis may be more straightforward to cluster with high confidence than some of the more problematic samples/cases (like spatial niches with complex mixtures of cell types) where single-cell resolution would be of very high interest.

These points are discussed in more detail below:

Shape sizes, normalization and data completeness

The concept of "single shapes" is not very articulated in the text and need to be better explained for reading clarity. It is unclear how cell contours are converted to shapes for laser cutting, except that shape sizes could be quite variable ($30 \mu\text{m}^2$ - $300 \mu\text{m}^2$). Was there any exclusion criteria (for example shapes including a nuclei or not)? How was the membrane inclusion or handling of shape outlines at cell boundaries (was there any offset on the contours)? These criteria might be quite critical both to generate solid data (for example to avoid contamination of the proteomes from neighbouring cells) and to interpret the data accurately.

Authors cut shapes from signal hepatocytes to achieve single cell resolution. It should however be noted that these cells are quite large compared to other cell types like lymphocytes.

Based on the reporting summary and the Methods section, for the single-shape samples, 8.9% of the samples were discarded (dropout) due to the number of proteins being lower than 806 (corresponding to median - 1.5 SD). Could the authors comment on what causes dropouts (i.e. is it related to the ROI selection or other technical factors)? What are the failure modes?

Supplementary Fig. S3d plots the data completeness, but there isn't a clear definition of what is meant by this. In the Methods it is mentioned that 100% completeness refers to a set of 175 proteins quantified across all samples. Hence, without more explanation, I am confused what this plot is showing.

A median normalization seems to be made based on these 175 proteins to correct the protein numbers across shapes of different sizes. This point is only mentioned briefly in the Methods section, but could be key for interpretation of the results obtained by this method. Hence it merits a more in depth explanation in the main text.

What are the sources of the high variance shown in Fig. 2a for scDVP?

Contribution of the different factors to the sensitivity

Although the key improvement for high sensitivity is the optimized sample processing workflow, relatively little data is provided regarding that optimization. More information and data about the relative contribution of the new parameters introduced should be included (for example: how much is use of DDM helping vs. the improvement obtained by the low flow gradient? What were the key contributors of the sensitivity improvement in the new protocol).

Could the authors provide a comparison of the total sizes of the pooled tissue pieces in the previous DVP paper vs. the single shape sizes in this manuscript?

If I have not missed it, I did not see information about the amount of protein extracted from the sample and injected to the mass spec (10 ng of the reference, but not sure what the rough estimate of the protein amount was for the single shapes).

Results

There should be more data and discussion about the kind of biases to expect in the data. Ideally data should be provided to show the distribution of different protein classes in the obtained proteomes to answer questions like:

How is the representation of the different protein classes such as membrane/secretory proteins?

How is the representation of the proteins from different subcellular localizations/organelles?

What is the observed detection bias based on sizes of proteins?

How would the intensities correspond to rough copy numbers of proteins?

Supplementary Fig. S3c shows ranking of the proteins by median intensity. Another plot that would be helpful for this figure is how bulk rare proteins they detect?

Supplementary Fig. 1 shows the results from 5 shape proteomes, which if I understand correctly are obtained with the original protocol (this needs to be clarified in the legend). Could authors also provide the PCA results for the single shape proteomes for comparison? For the titration experiment, it is not discussed at what shape number, it becomes not possible to resolve the liver zonation. This is important because it provides a benchmark for the sensitivity needed for resolving the clusters.

Related to the above point, although I understand the point of performing the proof of principle on liver zonation, when I consider the potential of the method, this application focusing on hepatocyte proteomes appears to be not a big enough challenge, i.e. a rather low bar, especially if relatively similar zonation profiles could have been resolved well enough with lower sensitivity data (as queried above). I am curious to know how the scDVP method would perform for other potentially more challenging use cases like cell typing and cell state assignment in complex tissues with many different cell types and how it will compare to multiplexed antibody-based spatial proteomics, spatial transcriptomics and single-cell RNA-Seq. For a comprehensive benchmark, a commonly used complex benchmarking tissue where in

addition to scRNA-Seq more extensive spatial omics data (spatial transcriptomics and antibody based spatial proteomics) would be available in atlases, could have provided more insights about the power of the method for more general single-cell spatial proteomics applications. Still, gaining metabolic insights based on the proteome profiles, as done here, is an important demonstration of the strength of the method and is well executed.

The prediction model is a nice addition to demonstrate the future potential of the method. Again, it would be wonderful to see the performance on a more complex case with higher proteome variation (such as niches or cancer samples).

Line 201: 17 features that are included seem to be only based on staining intensities. Were other morphological features (such as cell shape, nuclear size/shape etc.) not useful to include in the feature set or were these not extractable from the images?

In addition to the comparison to scRNA-Seq data, it would be helpful to see a validation in the spatial context (ideally by immunofluorescence). For example intensity distributions of some of the proteins that do not correlate well with the RNA data in supplementary Fig. 6c-d, should be ideally validated in the tissues by performing immunofluorescence for these targets in a consecutive section or replicate section.

In the imaging data provided by the authors, I can see the stitched images and the single-cell crops of phalloidin staining with the contours overlaid on them. Are these contours the segmented cells or the cut out shapes? Some of these look like holes in the tissue, but it is hard to tell for sure from the crops with only phalloidin staining. Can the authors provide stitched images with the cut out shapes directly marked on the big tissue image, so readers can see the cut and analyzed regions more clearly?

Based on the Methods only CFP staining (corresponding to phalloidin - although this is not explicitly written in the manuscript) was used for hepatocyte identification and segmentation, with a rather low confidence. Looking at the provided raw imaging data, this channel also seem to have bleaching artefacts. Were those regions manually excluded from analysis?

Discussion

Potential for post-translational modification detection could be discussed.

A discussion of limitations of the approach and potential for applications on complex tissues, cells of smaller sizes would be essential.

Scalability, throughput and cost of the method should be discussed.

What are the advantages and disadvantages of the mDIA approach? Can more than two single shapes be pooled together?

Considering the advantages and limitations of the method, what are the best use cases, what applications would require substantial future development?

Line 256, what is meant by "modular"? More elaboration on compatibility with other methods would be needed - what aspects of sample preparation render the sample inaccessible/accessible to other methods?

Comments on the potential importance of sample preparation methods (fresh frozen vs. FFPE) would be helpful.

Code availability

Will the authors provide a free compiled version of BIAS to reproduce their work?

Minor points:

No legends are provided for the Supplementary Tables as far as I can see.

Fig. 2a statistical descriptors are missing in the legend (boxplot elements, error, sample size are not defined).

Supplementary Fig. S3 panel legends are mixed up.

Fig. 1 and Fig. 4 legends - channel colors are not described for the overlay images.

Reviewer #3:

Remarks to the Author:

In the current manuscript Rosenberger et al. extend on their recent Deep Visual Proteomics Nature Biotechnology paper (ref 4.) and describe a single cell proteomics study on hepatocytes directly extracted from mouse livers. As these cells are rather big (20-30 μm) and their slices about 10 μm they are in a similar regime. The novelty of the paper they claim is that it extends single cell proteomics from cultured cells to cells taken out directly of frozen tissues. The methods they use for the analysis are identical as in ref 2 and 4, and thus previously described. The depth in proteome they report is good, but also as expected as these cells should not behave differently than cultured cells. They take out cells of the tissue in a directional manner to see whether cells taken out from different zones would exhibit proteome signatures. Using isotope labeling strategies, based on dimethyl labeling (a technique out for more than a decade!) they claim to detect more than 1,700 proteins per single shape (and up to 2,700) despite working from sections that were fixed, stained, imaged and laser dissected. They further state that the proteomics data from the single shapes correctly and accurately recapitulate the mouse liver physiology by direction, extent and spatial organization of zonation and compare their data with scRNAseq data to report great agreement.

The paper is written as a clear statement that everything was great and worked well. Maybe that is not that surprising as the work is a great achievement but a sort of natural extension of the work described by the authors in ref 2 and 4. Still, this does not make it again a great piece of work of the Mann laboratories. I do not like some of the claims made by the authors, of which I exemplify the term in the introduction: "true scProteomics" sort of claiming that others don't do true single cell work. There are more statements like this, that the authors really don't need

There are a few questions that came to mind reading the paper. The hepatocytes in the liver are known to secrete quite heavily proteins (for instance also into the blood stream). Moreover, in between the hepatocytes there may indeed also be blood. In the paper I do not find any discussion about such possible artefacts.

In discussing the differences of the shape proteomes in the paper, it becomes apparent that the major finding is that mitochondrial proteins define the signature. Do we really need proteomics to make that observation or would any mito_marker provide the same observation, and is this something they still could do. Most obvious this observation is presented in Suppl Fig 2b.

Although suppl fig 1d is color full it provides not any biological information, and moreover the clusters do not separate that well. Question is whether again the mitochondrial proteins do make the distinction?

The paper would benefit from a discussion about some of the current weaknesses and challenges, it reads now like everything worked perfect and is easy, while it is quite clear that only a handful labs in the world can do this. It would therefore also be interesting to discuss how other labs could adopt these approaches, as I guess that is a main reason to publish it in Nature Methods. A discussion about how much time/money this will costs and how it probably can be made easier for others would be beneficial.

In summary, this work is again a technical major achievement, but also a natural next steps following two recent publications from the same group. There is relatively little new biology as most of the story is build upon two well-known markers, Asl and Cyp2e1. Therefore, the proteomics community will very much appreciate this work, but liver physiologists will not yet become that excited about single cell proteomics.

Author Rebuttal to Initial comments

2023-05-17

Point-by-point answer to reviewers of the manuscript

Spatial single-cell mass spectrometry defines zonation of the hepatocyte proteome by Florian A. Rosenberger, Marvin Thielert, Maximilian T. Strauss, Constantin Amma, Sophia C. Mädler, Lisa Schweizer, Andreas Metousis, Patricia Skowronek, Maria Wahle, Janine Gote-Schniering, Anna Semenova, Herbert B. Schiller, Edwin Rodriguez, Thierry M. Nordmann, Andreas Mund and Matthias Mann

We would like to thank the reviewers for their constructive input, which has helped us to improve the manuscript. We are pleased that all three reviewers appreciate the potential of single-cell proteomics in intact tissues and that they appear to support publication after the requested modifications. In the revised manuscript, we have answered all questions and implemented the suggested changes in the manuscript as indicated in our detailed answers below. In summary, we made the following overall changes and additions:

All three reviewers requested increased accessibility to the manuscript's data. To ensure maximum transparency, we will make the associated R and Python code available on GitHub under a permissive Apache license. To facilitate pre-publication access for the reviewers, we have established a GitHub user account specifically for the private repository. To gain entry to the repository, the reviewers must log in to Github.com and authenticate using the provided credentials.

[Redacted]

The provided code will produce all data-containing figures of the manuscript. In addition, we now submit a rich set of supplementary materials including processed raw data, meta data, principal components, and statistics. Depending on *Nature Methods* preferences, we will also upload a zipped version to their website (120 Mbytes).

1. We provide additional methodological details in the Material and Methods as requested by the reviewers. We also add further information on limitations and prospects of the method in the Discussion part, specifically scalability, and extension to post-translational modifications.
2. In response to reviewer 1, we have re-analyzed much of our data, switching from principal component-guided sample sorting to truly spatial sorting throughout most of figure 3 and associated supplementary figures. We have further simplified the binning strategy, and now provide figures and data based on twenty spatial bins instead of eight PC1-guided bins, corresponding to the approximate number of hepatocytes along the liver zonation axis. Whenever reasonable we now show data from single shape measurements such as in Figure 3d.
3. This further allowed us to address suggestions by reviewer 2 by incorporating further biology. Specifically, we extended our analysis of the spatial proteome to changes in organellar sub-proteomes – while keeping in mind that the paper is mainly methodological in scope. The new main figures are Fig. 3g and 3h.
4. As requested especially by reviewer 3, we toned down the language throughout the manuscript and specifically rephrased parts of the introduction and discussion to make clear that single cell proteomics is not a development by a single group, but rather one branch of a global effort in the proteomics community.

Please refer to the answers below for more in-depth discussion. Original comments by reviewers are in black. Answers by the authors are in blue. For the exact changes to the text, please see the sections in red in the revised manuscript (and the ‘tracked changes’).

Reviewer #1

In the current study, Rosenberger et al. used an approach of combined imaging, laser capture

microdissection and MS to characterize the spatial proteome of small cellular fraction of murine hepatocytes. The authors modified their previously published protocol on single cells proteomics applied on suspended cells to allow single-cell proteomics evaluation with spatial context. The authors combined several staining markers to identify the liver lobule architecture. They first assessed what is the minimum area to collect to achieve significant and reproducible proteome measurement. Next, they collected 455 single shape samples and binned them to 8 zones, and found that the proteome measurements of these shapes recapitulate liver lobule biology of previously zoned genes. Finally, the authors developed an AI tool to predict single shape proteome based on microscopic images. This is the first demonstration of single cell proteomics in an intact mammalian tissue, an impressive achievement that warrants publication in *Nature Methods*.

We thank the reviewer for the thoughtful evaluation of our study. We appreciate the recognition of our work in developing an approach for single-cell proteomics with spatial context in intact mammalian tissue and that they believe it warrants publication in *Nature Methods*.

Several points should be addressed in the revision:

1. In general the data supplied do not enable easily reconstructing the analysis and applying new analyses. The authors should provide:

- a) Their reconstructed pc coordinates for each cell.
- b) The measured distance to the portal veins and central veins for each cell (shown in figure 3b).

We agree with the reviewer that readers should be able to easily reproduce our analyses from minimally processed raw data. To this end, we have implemented two measures: (a) With the revised manuscript, the reviewers also receive the data underlying each figure in tabular form. This also follows the recommendations of *Nature Methods*. (b) We now make all our R and Python code, as well as minimally processed raw (after DIA-NN

and RefQuant) and meta data available on GitHub (and Nature Methods web site in full or with links to the data, if desired by them). By executing the file ‘_Top_code.R’, the figure-specific R files will execute (also provided) which will then reproduce all data-containing figures that depend on R code. For the Python scripts, we used Jupyter Notebook and these are also deposited. It should be noted that the microscopy images have to be downloaded separately from the BioImage Archive as these files are too large for GitHub. The BioImages credentials are given in the *Data Availability* section of the manuscript. We are confident that this fully transparent documentation will allow readers to conveniently work with the data at their convenience.

The GitHub folders are password-protected prior to publication, but the reviewers can access them via the credentials

[Redacted]

c) the nuclear size, as estimated from the catapulted cellular image. This will enable easy analysis of protein zonation, as well as protein changes between hepatocytes of different ploidy classes.

The issue with nuclear size in these 10 um slice sections is that the nucleus in any given cell could be present across its middle, top or bottom or missing entirely. As such we think that providing the nuclear size is not meaningful. However, we did go through all images manually and now provide the number of nuclei per image in Supplementary Table S3, allowing spatial comparisons of mono- and binucleated hepatocyte proteomes – an interesting new analysis that is detailed just below.

2. Related to the above point, hepatocytes at the analyzed ages are either diploid, tetraploid or octoploid, while the authors analyzes thin tissue sections, nuclear diameter can still be used to at least estimated ploidy, are there differences between the proteomes of hepatocytes of different ploidy that reside within the same zone? Hepatocyte ploidy impacts gene expression in an only marginal manner (<https://pubmed.ncbi.nlm.nih.gov/34253736/>), it would be interesting to explore the impact on the hepatocyte proteome, given the new tool presented.

We agree that this is certainly an exciting question. We now use our manual annotation of mono- (n = 149) and binucleated (n = 73) cells, mentioned just above, and the absolute spatial coordinates to correct for liver zonation. In line with the previous reports mentioned by the reviewer, we also failed to detect any regulated proteins between the two groups in our data, (**Figure R1A** and **R1B**). However, a few proteins were exclusively detected in binucleated cells (17 proteins) or mononucleated cells (55 proteins; **Figure R1C**). The candidates in mononucleated cells with the highest detection count are shown in **Figure R1D**. This in itself points to interesting biology, but would require substantial molecular follow-up which is beyond the scope of this paper. However, we now make these annotation available so that readers can address this question on their own (for example, using the provided R scripts).

Figure R1: Absence of clear proteomic differences mononucleated (n = 149) versus binucleated (n = 73) cells and proteins exclusively detected in them. (A) PCA plot colour-coded by mono-versus binucleated cells in PC1 versus PC2, and **(B)** main driver proteins in PC4 versus PC6. **(C)** Unique and shared proteins. The number on top of the bar is total number of proteins in the respective group. **(D)** Top nine candidates found exclusively in mononucleated cells. Each dot corresponds to one measured shape. 3.

Figure 1a middle panel suggests that delta4 and delta8 pertain to mono or bi-nucleated hepatocytes, is this the case? Please explain better.

We did not imply any biological meaning in this illustration, and have now changed this to two mononucleated cells.

4. Zonation reconstruction – why do the authors use the PC1 rather than the absolute distances to the portal/central veins? It seems these were recorded for each cell.

The reviewer raises a great point that was also picked up by many readers of our pre-print. We do agree that the manuscript will profit from choosing absolute distance for spatial reconstruction instead of PC1. To demonstrate the power of our spatial approach, we harmonized most panels to use absolute distances (Figure 3, Supplementary Figure S6 – S9). Please note that we had to remove 50 single-cell measurements, in which distance to the next portal/central veins could not be reconstructed for sure, e.g., in cases of a cryostat artifact (documented in Supplementary Table S3). In the revised manuscript, we are still using PC1-guided clusters for whole-slide proteome reconstruction by machine learning to infer the spatial component, as that model must be informed by proteome and fluorescence values, whereas the spatial component should be predicted.

5. The authors compared their data to RNA zonation, please compare also to protein zonation obtained via spatial sorting in <https://www.nature.com/articles/s42255-019-0109-9>. It seems like the dynamic range in the current single cell dataset is lower than the spatial sorting study. Is there some background intensity that should be removed from the proteome? In addition, while zonation trends between the two studies nicely match, absolute levels of proteins do not, can the authors provide some methodological explanation to such discordances?

This is a very helpful suggestion. We added the comparison to this paper by Ben-Moshe *et al.*, finding good alignment between our and their data (Supplementary Fig. S8). Notably, the protein/protein correlation is slightly lower (Pearson's R of 0.87) than protein/RNA comparing to Halpern *et al.* (Pearson's R of 0.97), which initially surprised us. However, upon closer inspection, we noticed that the Moshe *et al.* study had not picked the midlobular zone (their Fig. 3b quadrant in the low left). This manifests in their data as a gap in the linear changes of protein intensity and also lowers the correlation to our data.

Please also note, that the two proteomics studies use a fundamentally different approach. While we use absolute distance values of a single cell to portal/central vein, Ben-Moshe *et al.* used fluorescence-activated cell sorting and inferred this information from fluorescence of two membrane markers, CD73 and Cdh1. Thus some differences would be expected from this alone. Furthermore, Moshe *et al.* analyzed more than 100,000 cells at a time whereas we analyzed about 1/3rd of a cell, which would be expected to reduce the observed dynamic range of protein expression.

6. In general, aspects of data analysis description in the paper is missing, would be important to elaborate on the pre-processing.

We agree, and now divide the data processing part of the Methods section into three headings and elaborated more in each section: (a) pre-processing with RefQuant, (b) normalization in R, (c) figure generation. Having all code on GitHub using the minimally processed data (after RefQuant) will hopefully make it easy to look under the hood concerning data (pre-) processing.

7. The authors' data nicely show mid-lobule zonation of genes such as Hamp2 and Cyp8b1, previously shown to have non-monotonic zonation patterns (<https://www.nature.com/articles/nature21065>). I would mention this in the text.

The reason we did not draw attention to this is that Hamp2 is not covered very well, and the mid-lobule peak of Cyp8b1 is not striking. That is why we prefer to not draw biological conclusions from this.

8. "Only contours between 30 μm^2 and 300 μm^2 – in 13w old mice is there such a diversity in cells size? How did you make sure you sample only hepatocytes?"

We thank the reviewer for picking this up – we got this comment also from readers of our preview. In fact, there was a technical mistake due to a missing conversion factor from pixels to micrometers. We have corrected this in the *Methods* section, and now provide updated values in all figures. As a result we realized that we had included cells that were

too large. In the revision, the size cut-off in our R script is now more stringent to match the BIAS software cut-off of 135 - 1350 μm^2 (Figure R2 and Supplementary Fig. S3c). This additional filtering excluded eight very large cells, resulting in a total of 399 hepatocytes and six validation endothelial structures. This does not change any of the conclusions. The new method's section text is now:

“Samples were excluded if the number of detected proteins was below 1.5 or above 3 standard deviations from the sample identification median, or within [792, 3312] identified proteins, resulting into a dropout of 9.2% (42 of 459 samples). Four samples were removed due to their outlier position on the PCA, see Supplementary Table S4. Eight samples were removed due to cell sizes larger than the BIAS cutoff of 1350 μm^2 . This resulted in 405 included samples, of which 399 were hepatocytes and 6 endothelial structures for validation.”

Liver cell types other than hepatocytes, e.g., endothelial cells or Kupffer cells, are much smaller and have a different shape and morphology compared to hepatocytes. We removed these both by the size cut-off mentioned above, and the AI-guided image segmentation. That we only sampled hepatocytes is also evident by the outlier position of endothelial cells in the PCA (Supplementary Fig. S4e). Note that this size selection strategy is unique to liver, and additional cell type markers will have to be used in most other tissues.

Prompted by the reviewer's comment, we have further included a new supplementary figure S3c depicting the size distribution of all hepatocytes, and separately mono- and

Figure R2: Size distribution of hepatocytes included in this study. (A) The numbers on top of the vertical lines are mean sizes as μm^2 of the group indicated by the respective colour (also new Supplementary Fig. S3c). (B) The category 'not classified' refers to cells which could not be confidently assigned to either mono- or binucleated cells. Boxplots depict median area as thick line with first and third quartile as box, whiskers are ± 1.5 interquartile range, outliers are dots.

binucleated cells (**Figure R2**). Their sizes fit the literature values for hepatocytes (about $600 \mu\text{m}^2$), and binucleated cells are about 28% larger in diameter or 63% in volume as expected. The boxplot below also answers the second question of the reviewer, showing that the majority of cells were in a narrow size range, and not atypically large or small (due to the sectioning plane).

Reviewer #2

Rosenberger et al., provides an update of the Deep Visual Proteomics method that was previously developed by the Mann lab. This technical update offers increased sensitivity, which allows the authors to achieve single cell resolution from. Main elements of this critical improvement were: use of surfactant n-Dodecyl- β -D-maltoside (DDM) to maximize peptide recovery, lowering the chromatographic flow rate for increased ionization efficiency, achieving higher chromatographic resolution with zero dead volume columns, and addition of a labeled reference channel for multiplexed data-independent acquisition (DIA) that decouples identification and quantification.

The improved method is verified by a demonstration that captures the spatial zonation of the hepatocyte proteome in the mouse liver tissue along the portal-central vein axis, including a comparison to previously published single cell RNA sequencing data. Using the acquired spatial proteomics data, the authors also train an AI model to predict proteome classes from imaged parameters.

The current work can be seen as an exciting extension of the Mund et al., 2022, Nature Biotechnology, which was a big leap towards untargeted spatial proteomics. It is mainly driven by optimization of the sample preparation workflow to increase the sensitivity and hence allow proteomic analysis from lower inputs corresponding roughly to single hepatocytes. I think the promise of the method for realizing untargeted spatial proteomics from fixed tissue samples is very high and I am happy to see the DVP method reaching a higher performance. Overall, I think the scDVP is definitely the cutting-edge of this class of technologies and this first attempt to utilize AI for predicting proteome classes from images is also quite interesting.

Having said that, major general weaknesses I see with the manuscript are: Although it is a technical update over the first DVP implementation, some of the relevant technical details about the technical improvements, critical considerations for assessment of the results and potential limitations of the method are glossed over. The implementation on the liver zonation, although looking very sound, may not be the best benchmarking/demonstration case

to highlight the capabilities of the method as proteome profiles of hepatocytes (being rather large cells) with clear metabolic differences along the chosen spatial axis may be more straightforward to cluster with high confidence than some of the more problematic samples/cases (like spatial niches with complex mixtures of cell types) where single-cell resolution would be of very high interest.

We thank reviewer for these positive and insightful comments on our work, which clearly reflect deep expertise in the subject matter. We appreciate the positive feedback on the Deep Visual Proteomics method and the technical extensions made upon it here and that the reviewer found the demonstration of capturing the spatial zonation of the hepatocyte proteome in the mouse liver tissue to be sound and exciting. In response to the constructive criticism regarding the technical details of the method, the assessment of the results, and the limitations of the approach, we have added substantial detail to the revision. Please find our responses below.

These points are discussed in more detail below:

Shape sizes, normalization and data completeness

The concept of "single shapes" is not very articulated in the text and need to be better explained for reading clarity. It is unclear how cell contours are converted to shapes for laser cutting, except that shape sizes could be quite variable ($30 \mu\text{m}^2$ - $300 \mu\text{m}^2$). Was there any exclusion criteria (for example shapes including a nuclei or not)? How was the membrane inclusion or handling of shape outlines at cell boundaries (was there any offset on the contours)? These criteria might be quite critical both to generate solid data (for example to avoid contamination of the proteomes from neighbouring cells) and to interpret the data accurately.

We agree that the concept of a "single shape" in DVP is more complex than it first appears and that it requires elaboration. We added technical notes to the Main text and *Methods* section for clarification.

In summary:

Cells were selected by a pre-trained AI-guided segmentation model, and size was the only cut-off chosen, as hepatocytes are considerably larger than all other cell types in the liver, e.g. endothelial cells or Kupffer cells. This strategy is unique for the liver, and different biological questions might need additional cell-type specific markers.

For several reasons, we did not choose the number of nuclei as an exclusion criterion. Firstly, the adult mouse liver is characterized by a very high degree of binucleation, and thus a nuclear selection criterion might bias towards specific populations. Secondly, a ten-micrometer section randomly cuts through cells with a certain proportion of non-nucleus single cell shapes remaining. We deem these equally important in a robust spatial proteomics workflow and do not exclude them.

We chose to cut shapes without additional offset, and now specify this in the Methods section. We agree with the reviewer's reasoning that contamination from surrounding material could impact the data, and to minimize this, we set the laser cut at the cell membrane. To address this important point, we provide a new panel g in Figure 3, which shows the membrane proteome is not affected by either laser accuracy or potential laser burning artifacts. This is apparent because the plasma membrane proteome is equally well represented in our data compared to the bulk liver library used in our study, and even the subcellular distribution of proteins in the Human Protein Atlas (**Figure R3**).

Figure R3 – Proportion of proteins assigned to the indicated subcellular compartment within the Human Protein Atlas (HPA) dataset (left) or our scDVP dataset (right). The category “Other” describes ambiguous mappings with multiple subcellular locations and unclear locations. See also Figure 3g for the comparison with bulk liver data.

Authors cut shapes from signal hepatocytes to achieve single cell resolution. It should however be noted that these cells are quite large compared to other cell types like lymphocytes.

Certainly, and this is why we think hepatocytes are a great benchmarking model for single-cell proteomics. We have included a sentence on this in the Introduction:

“[Given hepatocyte sizes of 20-30 μm , one shape cut from a 10 μm section corresponds to a third or half of a hepatocyte,] equivalent to about 250 ng of total protein input per shape or a complete HeLa cell equivalent.”

We believe that further ongoing technical improvements will make it possible to obtain similarly deep spatial proteomes of smaller cells like lymphocytes. In the context of this study, it should be noted that the included protein mass analyzed is equal to or is less than one HeLa cell, which is used as a benchmark in many other single cell proteomics papers.

Based on the reporting summary and the Methods section, for the single-shape samples, 8.9% of the samples were discarded (dropout) due to the number of proteins being lower than 806

(corresponding to median - 1.5 SD). Could the authors comment on what causes dropouts (i.e. is it related to the ROI selection or other technical factors)? What are the failure modes?

A discussion on failure modes is certainly of interest for the readership of *Nature Methods*. We have included the dropped-out shapes in the revised figure S3d showing that these were overall particularly concentrated in small shapes. In addition, we had previously observed evaporation from some of the affected samples during high-temperature steps. In the methods section, we now write:

“During each incubation step, plates were tightly sealed with two stacked aluminum lids (Thermo Fisher Scientific, AB0626) to avoid evaporation.”

While setting up the method, we found that a ‘wind shield’ inside the laser microdissection microscope substantially increased the number of successfully measured cells. This shield is commercially available from Leica, the microscope’s manufacturer, and comes with the instrument. We added this information to the Methods section as

“A ‘wind shield’ plate was used on top of the sample stage to ensure proper sorting.”

Supplementary Fig. S3d plots the data completeness, but there isn’t a clear definition of what is meant by this. In the Methods it is mentioned that 100% completeness refers to a set of 175 proteins quantified across all samples. Hence, without more explanation, I am confused what this plot is showing.

In the figure legend of Figure S3, we added *“Data completeness is defined as percentage of samples across all samples in which a particular protein was quantified in”*.

A median normalization seems to be made based on these 175 proteins to correct the protein numbers across shapes of different sizes. This point is only mentioned briefly in the Methods

section, but could be key for interpretation of the results obtained by this method. Hence it merits a more in depth explanation in the main text.

We fully agree, and also accept that this is a field where major improvements can still be made in the future. We have added additional information to the Methods section. Furthermore, the complete R and Python code required to reproduce the figures of our study are available on GitHub, and can be accessed by reviewers prior to publication via

[Redacted]

What are the sources of the high variance shown in Fig. 2a for scDVP?

The main reason for the high variance of the number of protein IDs in Figure 2a is the size differences of excised shapes, as larger shapes will result in more quantified proteins, as we are operating at the threshold of MS sensitivity. This is shown in supplementary Figure S3d. To exemplify this, we have grouped cells by size into four equidistant bins in the boxplot below, showing that cells of similar sizes result in a similar number of protein IDs (**Figure R4**). This illustrates that single-cell proteomics and especially single-cell spatial proteomics pushes the envelope of proteomics capabilities, with further technological improvements directly translating into enhanced biological potential.

Figure R4– Number of protein IDs, binned by size of cut single shapes. Boxplots depict median IDs as thick line with first and third quartile as box, whiskers are +/- 1.5 interquartile range, outliers are dots.

Contribution of the different factors to the sensitivity

Although the key improvement for high sensitivity is the optimized sample processing workflow, relatively little data is provided regarding that optimization. More information and data about the relative contribution of the new parameters introduced should be included (for example: how much is use of DDM helping vs. the improvement obtained by the low flow gradient? What were the key contributors of the sensitivity improvement in the new protocol).

We agree that readers could be interested in learning in more detail where the sensitivity improvements come from. That said, the contribution of this paper is the demonstration of the feasibility of single-shape proteomics in tissues and the fact that biological results can be drawn from this. Analytical chemistry considerations on the triple labeling workflow, including the DDM boost, are either known in the literature or described in more depth in our mDIA paper by Thielert *et al.*, specifically the effect of the reference

channel (referenced in our manuscript *Robust dimethyl-based multiplex-DIA workflow doubles single-cell proteome depth via a reference channel*. Thielert M, et al. *bioRxiv* 2022.12.02.518917; doi: <https://doi.org/10.1101/2022.12.02.518917>).

Could the authors provide a comparison of the total sizes of the pooled tissue pieces in the previous DVP paper vs. the single shape sizes in this manuscript?

In the DVP paper by Mund *et al.*, approximately 40,000 – 50,000 μm^2 (700 contours x 60-70 μm^2) were used per sample, while herein we use 135-1350 μm^2 .

If I have not missed it, I did not see information about the amount of protein extracted from the sample and injected to the mass spec (10 ng of the reference, but not sure what the rough estimate of the protein amount was for the single shapes).

As we inject the proteome of a whole shape without quantifying before, we do not know the exact amount of protein per sample. One hepatocyte has a diameter of approximately 20 to 30 μm , and a protein content of about 700 pg. As our slices are 10 μm thick, we retrieve one third to half of a cell, corresponding to 230 – 350 pg. As already mentioned, this is very similar to the amount of protein in one HeLa cell, which is a typical study object in the field of single-cell proteomics. Notably, our shapes undergo the whole scDVP procedure with fixation, staining, and cutting, thus likely accounting for the ~25 % lower numbers in protein IDs compared to our technical mDIA manuscript (referenced above).

Results

There should be more data and discussion about the kind of biases to expect in the data. Ideally data should be provided to show the distribution of different protein classes in the obtained proteomes to answer questions like:

How is the representation of the different protein classes such as membrane/secretory proteins?

How is the representation of the proteins from different subcellular localizations/organelles?

What is the observed detection bias based on sizes of proteins? How would the intensities correspond to rough copy numbers of proteins?

We thank the reviewer for bringing up this interesting question, which prompted us to do additional analyses of our data. We now plot the coverage of proteins from different subcellular localizations in the new panel **Figure 3g**, and also below in **Figure R4**. The comparison data was taken from the subcellular Protein Atlas (Thul *et al.* A subcellular map of the human proteome. *Science* 356, eaal3321(2017)). We cross-mapped the human proteins to our mouse scDVP data by the DIOPT Ortholog Finder (Hu *et al.* An integrative approach to ortholog prediction for disease-focused and other functional studies. *BMC Bioinformatics* (2011)), discarding proteins that did not have a clear hit. This resulted in localization data for 2,264 proteins out of our total of 3,588 (63.1%).

Reassuringly, plasma membrane proteins by identifications were represented to a similar extent in the scDVP dataset (2.7%) compared to the Human Protein Atlas data (2.8%). By signal intensity, the numbers were also similar in our scDVP dataset (10.7%) compared to the bulk liver proteome that we had generated (10.6%). Thus, the laser accuracy, or laser-induced artifacts did not distort the cell membrane proteome.

Supplementary Fig. S3c shows ranking of the proteins by median intensity. Another plot that would be helpful for this figure is how bulk rare proteins they detect?

We compared our data to a proteome of isolated human hepatocyte cells (hHEPs) which was part of one of our previous papers (Niu *et al.* Dynamic human liver proteome atlas reveals functional insights into disease pathways. *Mol Syst Biol.* 2022

Figure R5 - Comparison of scDVP proteomes with a bulk human hepatocyte proteome from Niu *et al.* (A) Intensities of scDVP mouse data versus hHEPs. Human to mouse protein orthologs were mapped with DIOPT Ortholog Finder (Hu *et al.* An integrative approach to ortholog prediction for disease-focused and other functional studies. *BMC Bioinformatics* (2011)). Only high scoring orthologs were kept, and non-matches were discarded. NAs were converted to 0s if detected in either hHEPs or scDVP data. The dotted line is the 45 degree intersection line. (B) Protein intensities in the scDVP dataset split by detection in hHEPs, and (C) vice versa for hHEPs proteins split by detection in scDVP dataset.

May;18(5):e10947.). This is plotted in **Figure R5**, showing that we do detect some low abundance hHEP proteins, but not at the same depth. Missing protein detections of the human sample when present in mouse are arbitrary (B), while there is a clear intensity bias for proteins uniquely detected in the human but not mouse sample.

Supplementary Fig. 1 shows the results from 5 shape proteomes, which if I understand correctly are obtained with the original protocol (this needs to be clarified in the legend). Could authors also provide the PCA results for the single shape proteomes for comparison?

Certainly, this is part of the underlying figure data that the reviewers are now receiving with this re-submission. Furthermore, the code is well documented on GitHub also for the five shape figures. We have now specified in the legend of supplementary Fig. S1 that these were measured with the original protocol.

For the titration experiment, it is not discussed at what shape number, it becomes not possible to resolve the liver zonation. This is important because it provides a benchmark for the sensitivity needed for resolving the clusters.

This is a great comment and very relevant question. We interpret this as how many adjacent cells on one spot can be cut (e.g. one shape, five shapes, hundred shapes), while still retaining enough information to retrieve known spatial biology. We cannot

Figure R6 (equal to the new suppl. Fig. 5): Concatenating cells rapidly leads to loss of biological information. (A) PCA after concatenation of cells that are closest spatial neighbours at given ratios (1:001- every single shape, 1:002 – combining two shapes, and so forth). (B) Interquartile range of principal components one to five after concatenation as in (A). (C) Variance explained by principal components one to four. See main manuscript text for further explanation.

address this experimentally by cutting dozens of single hepatocytes on one spot as this would result in significant contamination with non-hepatocyte cells such as endothelial cells and macrophages. (Unlike in our single-cell microdissection, these cells would be excised from the tissue together with the cells of interest).

Instead, we approached the reviewer's question by consecutively aggregating the data from hepatocytes with similar distance (true space) to portal and central vein. This analysis showed that PC1 remains resolved down to two bins (**Figure R6A**), which makes biological sense as portal to central vein in fact reflects the most extreme proteomics difference. Strikingly, however, information on additional components is lost early upon aggregation of as little as two neighboring shapes (**Figure R6B**), and PC1 unproportionally gains in information and variance explained (**Figure R6C**).

Related to the above point, although I understand the point of performing the proof of principle on liver zonation, when I consider the potential of the method, this application focusing on hepatocyte proteomes appears to be not a big enough challenge, i.e. a rather low bar, especially if relatively similar zonation profiles could have been resolved well enough with lower sensitivity data (as queried above). I am curious to know how the scDVP method would perform for other potentially more challenging use cases like cell typing and cell state assignment in complex tissues with many different cell types and how it will compare to multiplexed antibody-based spatial proteomics, spatial transcriptomics and single-cell RNA-Seq. For a comprehensive benchmark, a commonly used complex benchmarking tissue where in addition to scRNA-Seq more extensive spatial omics data (spatial transcriptomics and antibody based spatial proteomics) would be available in atlases, could have provided more insights about the power of the method for more general single-cell spatial proteomics applications. Still, gaining metabolic insights based on the proteome profiles, as done here, is an important demonstration of the strength of the method and is well executed.

The prediction model is a nice addition to demonstrate the future potential of the method. Again, it would be wonderful to see the performance on a more complex case with higher proteome variation (such as niches or cancer samples).

We share the excitement and enthusiasm to apply the scDVP method to other biological questions in which the tissue is more heterogenous and 'more convoluted' as in liver. We are also heartened by the reviewer's confidence in our experimental abilities. However, while we are actively working towards the goal of resolving i.e. tumor microenvironments and the immune cells within them, we believe that this is much beyond the scope of this already challenging work. Furthermore, it would risk to shift the focus expected of a *Nature Methods* article. That said, we absolutely agree that the method should be rolled out over different and challenging cases as soon as possible!

Line 201: 17 features that are included seem to be only based on staining intensities. Were other morphological features (such as cell shape, nuclear size/shape etc.) not useful to include in the feature set or were these not extractable from the images?

Of the 17 features we included, the majority were based on intensity, and only one was morphological ('Area'). For the classifier, the importance of Area was in the middle range of all features. While other morphological features could be extracted, and we believe they could be helpful, we decided not to do so. First, the current set of features can be easily validated and somewhat quantified visually so that they can be readily verified. Second, while the cohort of shapes is comparatively large in the single cell proteomics context, we still only have 302 samples for training and are limited in validating the model. By adding additional features, we also increase the possibility of overfitting.

Overall, we think that the current performance of the ML with mostly intensity-based features is remarkably good and that it clearly shows the potential of the technology. It certainly can be further improved, especially with increasing sample sizes.

In addition to the comparison to scRNA-Seq data, it would be helpful to see a validation in the spatial context (ideally by immunofluorescence). For example intensity distributions of some of the proteins that do not correlate well with the RNA data in supplementary Fig. 6c-d, should be ideally validated in the tissues by performing immunofluorescence for these targets in a consecutive section or replicate section.

Prompted by the reviewer's observation, we now evaluated this discrepancy by comparing our data with immunohistochemical staining from the Human Protein Atlas for three targets, in which proteomics and RNAseq data do not align well, namely Ngef, Pank4 and Spg20 (**Figure R7**). The scDVP data measured zonation along the periportal/peri-central axis correctly for all three. This suggests that the discrepancies noted by the reviewer – at least in some cases – reflect true differences between the transcriptome and proteome.

Figure R7 – Immunohistochemical stains of the indicated proteins. The legend above the images indicates the zonation profile. PV: Portal Vein, CV: Central Vein.

In the imaging data provided by the authors, I can see the stitched images and the single-cell crops of phalloidin staining with the contours overlaid on them. Are these contours the segmented cells or the cut out shapes? Some of these look like holes in the tissue, but it is hard to tell for sure from the crops with only phalloidin staining. Can the authors provide stitched images with the cut-out shapes directly marked on the big tissue image, so readers can see the cut and analyzed regions more clearly?

We are not entirely sure what the reviewer is asking. If they are referring to the information that we uploaded to BioRxiv, then these are the indeed cut-out shapes and not all of the segmented cells. Those can be visualized in the BIAS software (see below). A different version of images with cut out cells can be reproduced with the Python code provided on Github (DataPreparation.ipynb)

Based on the Methods only CFP staining (corresponding to phalloidin - although this is not explicitly written in the manuscript) was used for hepatocyte identification and segmentation, with a rather low confidence. Looking at the provided raw imaging data, this channel also seem to have bleaching artefacts. Were those regions manually excluded from analysis?

We did minimal curation of the list of included cells to avoid false segmentations, and now clarify this in the Methods section. However, it should be noted that the number of removed shapes was less than ten out of hundreds.

Discussion

Potential for post-translational modification detection could be discussed.

We added to the discussion:

“There have been advances in the quantification of post-translational modifications from ultra-low input material, such as from 1 μ g down to the material corresponding to single cell, for instance in the enrichment protocol μ Phos²⁰. In combination with scDVP, this holds promise for single cells, although the biological information in sc-phosphoproteomics data would currently be limited to a few high-abundance proteins with high modification stoichiometries. Subtle signaling events, such as the liver-dominant Wnt-signaling, will require additional technological developments for in-depth biological description of signaling in single cells by mass spectrometry.”

A discussion of limitations of the approach and potential for applications on complex tissues, cells of smaller sizes would be essential.

We agree and have added the following section to the discussion:

“Our results demonstrate that the central challenge of scDVP is the sensitivity of the overall workflow (Supp. Fig. 3d). Although we have here reduced the area required for laser microdissection by hundred-fold compared to our initial DVP report⁴, we note that one excised hepatocyte shape contains approximately ten times more protein than

the smallest cells of interest, such as typical resting lymphocytes¹⁹. While the required sensitivity is being developed, the original DVP approach using pooled cells of the same type is a powerful tool for this kind of problems. We also note recent success in drastically scaling down DVP for formalin-fixed and paraffin-embedded (FFPE) samples, which are readily available in many clinical settings²⁰.”

Scalability, throughput and cost of the method should be discussed.

What are the advantages and disadvantages of the mDIA approach? Can more than two single shapes be pooled together?

We agree that scalability, throughput, and cost are important factors to consider in any experimental method, and we appreciate the reviewer's suggestion to address these points. Regarding equipment, commercially available instruments such as the common Zeiss AxioScan 7 scanning microscope, Leica LMD7 laser microdissection microscope, and TimsTOF SCP or equivalents can be used. It is difficult and perhaps inappropriate, however, to note specific prices in a manuscript, as these vary widely between countries and institutions (i.e. academic and industry). Medical schools, for instance, will generally have pathology scanning microscopes already.

However, we are on firmer ground regarding the marginal costs of scDVP, which turn out to be quite manageable.

“We established the scDVP protocol to combine one reference channel with two single shapes (effective two-plex) and used a 40-samples per day chromatography method. This can be further scaled to five-plex (effective four-plex) and 80-samples per day, scaling to 320 shapes per day. Given the more stable core proteome compared to sc-transcriptomes² and the resulting lower required sample number, scDVP experiments encompassing a few hundred single shapes could be done in just a few days. Furthermore, because of the very low quantities and absence of proprietary reagents, marginal costs are extremely low.”

Considering the advantages and limitations of the method, what are the best use cases, what applications would require substantial future development?

We hope this is mostly covered by our previous answers, in summary the best use cases:

“scDVP can be a method of choice to map proteomic disturbances along gradients of, for instance, signaling factors, nutrients or gases, and in physiological settings that may create impediments for other omics methods, for instance in extracellular fibrotic scars.”

Line 256, what is meant by "modular"? More elaboration on compatibility with other methods would be needed - what aspects of sample preparation render the sample inaccessible/accessible to other methods?

What we mean is that we have made the pipeline easily adaptable by allowing every step to be upgraded or exchanged. Our system supports most imaging file formats, such as Zeiss CZI and Leica LIF, from different microscope vendors, in turn allowing the use of most scanning microscopes for image acquisition. The integration of open-source packages and provision of custom algorithms, including data access and manipulation, is possible through our Python interface, providing a generic way for data analysis. Our interface currently supports the Zeiss PALM MicroBeam and the Leica LMD6 and LMD7 laser microdissection microscopes, enabling single-cell isolation from both fresh and FFPE tissue samples for proteomic, RNA, or DNA analysis. Downstream bioinformatic analysis can be performed using the analysis pipeline preferred by the user.

To keep the main text short, we added some information to the discussion on what we think the most accessible steps are:

“The modular nature of scDVP, especially the open format from laser microdissection to 384-well plates for sample preparation, makes it widely applicable

and also compatible with other spatial omics technologies such as spatial transcriptomics, epigenomics²⁴ or multiplexed imaging.”

Comments on the potential importance of sample preparation methods (fresh frozen vs. FFPE) would be helpful.

Please see our answer above.

Code availability

Will the authors provide a free compiled version of BIAS to reproduce their work?

A free compiled version of BIAS is available from the BioImage Archive of the original DVP paper here:

<https://www.ebi.ac.uk/biostudies/bioimages/studies/S-BSST820>

We additionally deposited the BIAS pipelines needed to reproduce the work on our BioImage repository with the accession number S-BIAD596. These are already now accessible to reviewers with the link <https://www.ebi.ac.uk/biostudies/studies/S-BIAD596?key=428ca999-51e9-411c-b34b-21f06f5c7caf>

Minor points:

No legends are provided for the Supplementary Tables as far as I can see.

Thank you for spotting this. The table legends are now provided at the end of the manuscript, together with figure legends and supplementary figure legends in one block.

Fig. 2a statistical descriptors are missing in the legend (boxplot elements, error, sample size are not defined).

These are now added to the legends, including all supplementary figures where these were missing.

Supplementary Fig. S3 panel legends are mixed up.

Thank you for pointing this out! It is fixed now.

Fig. 1 and Fig. 4 legends - channel colors are not described for the overlay images.

Channel colors for the overlay images are now described in the figure legend.

Reviewer #3

In the current manuscript Rosenberger et al. extend on their recent Deep Visual Proteomics Nature Biotechnology paper (ref 4.) and describe a single cell proteomics study on hepatocytes directly extracted from mouse livers. As these cells are rather big (20-30 μm) and their slices about 10 μm they are in a similar regime. The novelty of the paper they claim is that it extends single cell proteomics from cultured cells to cells taken out directly of frozen tissues. The methods they use for the analysis are identical as in ref 2 and 4, and thus previously described. The depth in proteome they report is good, but also as expected as these cells should not behave differently than cultured cells. They take out cells of the tissue in a directional manner to see whether cells taken out from different zones would exhibit proteome signatures. Using isotope labeling strategies, based on dimethyl labeling (a technique out for more than a decade!) they claim to detect more than 1,700 proteins per single shape (and up to 2,700) despite working from sections that were fixed, stained, imaged and laser dissected. They further state that the proteomics data from the single shapes correctly and accurately recapitulate the mouse liver physiology by direction, extent and spatial organization of zonation and compare their data with scRNAseq data to report great agreement. The paper is written as a clear statement that everything was great and worked well. Maybe that is not that surprising as the work is a great achievement but a sort of natural extension of the work described by the authors in ref 2 and 4. Still, this does not make it again a great piece of work of the Mann laboratories. I do not like some of the claims made by the authors, of which I exemplify the term in the introduction: "true scProteomics" sort of claiming that others don't do true single cell work. There are more statements like this, that the authors really don't need.

We generally agree with the factual summary of the steps involved in scDVP and appreciate the reviewer's characterization that 'the work is a great achievement' (also repeated below). At the same time, the reviewer is at pains to point out that there are no major new or surprising developments and that this work is a natural extension of our previous work. We politely disagree with that assessment both regarding the technological advances necessary in making this method actually work and especially by the actual spatial output that far exceeded our initial expectations.

Regardless of these matters of opinion, we agree that we should indeed tone down our claims to novelty and make it crystal clear what we have developed here and what the significance of our biological results is. The other reviewers also noted some of these issues and we have commented on those above already. Overall, as in any scientific advance, we acknowledge that our success is based on the developments of many, many groups around the globe. As concrete example, we now omit mention of ‘true single cell proteomics’.

There are a few questions that came to mind reading the paper. The hepatocytes in the liver are known to secrete quite heavily proteins (for instance also into the blood stream). Moreover, in between the hepatocytes there may indeed also be blood. In the paper I do not find any discussion about such possible artefacts.

We thank the reviewer for bringing up this point. As a result of the prominent role of the liver in blood production and its high perfusion rate, bulk proteomics experiments often struggle with admixtures of the plasma proteome that are difficult to disentangle from the liver proteome. We reinspected our data and reassuringly (and expectedly in retrospect) we found that plasma proteins such as albumin are of medium abundance in our dataset (Fig. R8). Reassuringly, the erythrocyte proteins hemoglobin A, B and D were not in detected our dataset. As this observation adds to the paper, we added a sentence to the discussion:

“Our laser microdissection workflow successfully separated hepatocytes from surrounding material including blood remnants, which holds promise for smaller cell types in more complex tissue environments.”

We furthermore show in Figure R8 and supplementary material (fig. S3d) that blood proteins such as albumin are of medium abundance in our dataset.

Figure R8 (and Supplementary Fig. S3d) – Levels of common plasma proteins in the scDVP dataset.

In discussing the differences of the shape proteomes in the paper, it becomes apparent that the major finding is that mitochondrial proteins define the signature. Do we really need proteomics to make that observation or would any mito_marker provide the same observation, and is this something they still could do. Most obvious this observation is presented in Suppl Fig 2b.

We have reinspected our data and our analysis shows that the observed zonation effects are not just driven by the mitochondrial proteome. Furthermore, the mitochondrial proteome is very much affected by liver zonation; a fact apparent in our data and

Figure R9 – ANOVA effect size across eight spatially guided bins in dependence of subcellular localisation. Localisation data is from the Human Protein Atlas via ortholog mapping using DIOPT Ortholog Finder (included here: 486 proteins after filtering for significant proteins at FDR > 0.05). Only the top 10 subcellular localisations are shown.

independently highlighted in recent work by Kang *et al.* (<https://www.biorxiv.org/content/10.1101/2023.04.13.536717v1>). In fact, our data is particularly suitable to study mitochondrial function as these organelles are surprisingly well covered in our data sets (56.1% of the proteins in Mitocarta 3.0). Yet, the mitochondrial proteome is by far not the only contributor to liver zonation, which was also reported in the earlier RNAseq and FACS-based proteomics studies. **Figure R9** shows effect size of proteins (B) that significantly contribute to liver zonation, revealing that also the nucleolar proteome is highly zoned, with outliers across all subcellular compartment. This indicates that scDVP data is far richer than just the prominent mitochondrial zonation

drivers. As pointed out above, these new analyses are now incorporated in the revised manuscript (see also the new Suppl. Fig. 8).

Although suppl fig 1d is color full it provides not any biological information, and moreover the clusters do not separate that well. Question is whether again the mitochondrial proteins do make the distinction?

We added the main drivers of PCA separation to supplementary figure S1D (the five shapes proteomes), and three out of four of the top drivers of PC1 separation are not mitochondrial (Arg1, Aldh1a1, Acaa1b) and one has two isoforms (Cps1) in either the cytoplasmic or mitochondrial compartment. The clusters are based on unbiased k-means clustering that takes all principal components and not just the shown PC1 and PC2 into account. Therefore, overlapping clusters suggest that the data is complex and rich in biology that cannot just be explained by the variation of just a few proteins.

The paper would benefit from a discussion about some of the current weaknesses and challenges, it reads now like everything worked perfect and is easy, while it is quite clear that only a handful labs in the world can do this. It would therefore also be interesting to discuss how other labs could adopt these approaches, as I guess that is a main reason to publish it in Nature Methods. A discussion about how much time/money this will costs and how it probably can be made easier for others would be beneficial.

This was also pointed out by reviewer 2 and we agree that it is important to discuss the current limitations and challenges of our approach in order to provide a realistic view of the method. We have added several sections to the discussion addressing these points (see also above), highlighting the novel requirements of the method and the expertise required to perform it.

Regarding the cost and accessibility of the method, the equipment required for DVP is indeed currently expensive, requiring a state-of-the-art mass spectrometer and a laser capture microscope that may not be accessible to every laboratory (in common with much of high-end imaging or proteomics). Furthermore, the required expertise is

currently not wide spread although we see no reason that this could not change soon. In fact we have been contacted by many laboratories and are currently helping them to establish DVP, to which this will be just an add on. One of the major bottlenecks was to build a streamlined and user-friendly data processing pipeline, but once such a task is completed it is readily transferable. Specifically, we have adapted the BIAS software suite that combines image processing, deep learning-based image segmentation, feature extraction, and machine learning-based phenotype classification. This streamlined workflow can help reduce the complexity and cost of the method. We are working hard to make the entire workflow user-friendly and robust. That said, several steps in the pipeline can still be optimized. For example, we are making progress by developing a robust workflow for routine use of glass membrane slides, allowing precise extraction of tissue in combination with automated and multicolor immunofluorescence staining. We describe this work and a detailed protocol in Nordmann *et al.* here [biorxiv.org/content/10.1101/2023.02.20.529255v1](https://doi.org/10.1101/2023.02.20.529255v1)

In summary, this work is again a technical major achievement, but also a natural next step following two recent publications from the same group. There is relatively little new biology as most of the story is built upon two well-known markers, As1 and Cyp2e1. Therefore, the proteomics community will very much appreciate this work, but liver physiologists will not yet become that excited about single cell proteomics.

Thank you again for highlighting the 'technical major achievement'. Regarding the missing excitement of the biological community that is certainly not the impression we have gotten from the feedback of both academic and biotech laboratories. The liver community specifically was indeed quite excited, and we have received many requests to apply the method to cover liver disease states of their interest.

In fact, many proteins drive liver zonation (more than half are significantly different between portal and central vein at an FDR cut-off of 5%). Examples of new biology in our manuscript include the degradation of very-long chain fatty acids, which we localize to the peri-central regions, a finding that has not been known before. One of our favorite

figures is Fig. 3d, showing how Wnt signaling decreases exponentially and not linearly from the central to portal vein. This is in line with RNAseq data by Halpern *et al.*, but has never been shown on the proteomics level, even complete with absolute spatial coordinates.

Decision Letter, first revision:

Dear Matthias,

Thank you for submitting your revised manuscript "Spatial single-cell mass spectrometry defines zonation of the hepatocyte proteome" (NMEMH-A51109A). It has now been seen by the original referees and their comments are below. The reviewers find that the paper has improved in revision, and therefore we'll be happy in principle to publish it in Nature Methods, pending minor revisions to satisfy the referees' final requests and to comply with our editorial and formatting guidelines.

We request you to ensure that the code/program is available for use at the time of publication. Additionally, we recommend providing a supplementary protocol for this paper. Please see below.

SUPPLEMENTARY PROTOCOL

To help facilitate reproducibility and uptake of your method, we ask you to prepare a step-by-step Supplementary Protocol for the method described in this paper. We [encourage authors to share their step-by-step experimental protocols](https://www.nature.com/nature-research/editorial-policies/reporting-standards#protocols) on a protocol sharing platform of their choice and report the protocol DOI in the reference list. Nature Portfolio's Protocol Exchange is a free-to-use and open resource for protocols; protocols deposited in Protocol Exchange are citable and can be linked from the published article. More details can be found at www.nature.com/protocolexchange/about.

TRANSPARENT PEER REVIEW

Nature Methods offers a transparent peer review option for new original research manuscripts submitted from 17th February 2021. We encourage increased transparency in peer review by publishing the reviewer comments, author rebuttal letters and editorial decision letters if the authors agree. Such

peer review material is made available as a supplementary peer review file. Please state in the cover letter 'I wish to participate in transparent peer review' if you want to opt in, or 'I do not wish to participate in transparent peer review' if you don't. Failure to state your preference will result in delays in accepting your manuscript for publication.

ORCID

Sincerely,
Arunima

Arunima Singh, Ph.D.
Senior Editor
Nature Methods

Reviewer #1 (Remarks to the Author):

The authors have done a great job in addressing all of my comments and have significantly improved the paper.

Reviewer #2 (Remarks to the Author):

The authors responded to the comments of the reviewers in a comprehensive fashion and satisfactorily addressed the issues that were raised. I do not have any further major remarks.

A few minor thoughts are:

- It is nice to see the representation of proteins from different subcellular compartments. In Fig. R3, mitochondria and cytoplasm seem to be over represented whereas nucleoplasm proteins are underrepresented in the scDVP. Is there an explanation for that?
- Overall combining the subcellular location annotation and the morphology information could be quite interesting to do sanity checks and explore quantitiveness (for example, when a binucleated cell is captured, does the nuclear proteome distribution/contribution change?).
- The GitHub reviewer access still asks for email confirmation, so it seems this approach of sharing the code in advance does not work.

Reviewer #3 (Remarks to the Author):

The authors have extensively and adequately responded to my comments, as well as to the other reviewers. It is now a very nice piece of work

Print Email

Author Rebuttal, first revision:

Point-by-point answer to reviewers

Spatial single-cell mass spectrometry defines zonation of the hepatocyte proteome

by Florian A. Rosenberger, Marvin Thielert, Maximilian T. Strauss, Constantin Amma, Sophia C. Mädler, Lisa Schweizer, Andreas Metousis, Patricia Skowronek, Maria Wahle, Janine Gote-Schniering, Anna Semenova, Herbert B. Schiller, Edwin Rodriguez, Thierry M. Nordmann, Andreas Mund and Matthias Mann

We would like to thank all reviewers for re-evaluating the manuscript. We are pleased that all three reviewers approve of our changes and that they believe that the manuscript has significantly improved. There are no remaining comments from reviewers 1 and 3

and below, we address the few comments of reviewer #2. Original comments by reviewers are in black. Answers by the authors are in blue.

Reviewer #1:

Remarks to the Author:

The authors have done a great job in addressing all of my comments and have significantly improved the paper.

We thank the reviewer for the positive feedback, and the comments of the first round.

Reviewer #2:

Remarks to the Author:

The authors responded to the comments of the reviewers in a comprehensive fashion and satisfactorily addressed the issues that were raised. I do not have any further major remarks.

A few minor thoughts are:

- It is nice to see the representation of proteins from different subcellular compartments. In Fig. R3, mitochondria and cytoplasm seem to be over represented whereas nucleoplasm proteins are underrepresented in the scDVP. Is there an explanation for that?

There could be at least two reasons. Firstly, transcription factors that are assigned to the terms “nucleoplasm” and “nucleus” are of low abundance and are under-represented in single-cell proteomics data. (This is why we had included them as a quality control in Figure 2A). By the same token, our limited depth would be expected to lead to overrepresentation of mitochondria and cytoplasm, which are abundant. Secondly, the paper by Thul *et al.* which was the basis for the former figure R3 (see their Supplementary Table S7) provided subcellular localisation for more than 35,000 proteins

derived from transcriptomic data, clearly a very different data set than the proteomics data where in a situation like this, we would mainly measure one protein per protein coding gene. This might have led to some bias in the comparison, in conjunction with the fact that they used 22 different cell lines, not necessarily with rich mitochondria and highly abundant cytoplasmic enzymes, as is the case in our hepatocytes.

- Overall combining the subcellular location annotation and the morphology information could be quite interesting to do sanity checks and explore quantitiveness (for example, when a binucleated cell is captured, does the nuclear proteome distribution/contribution change?).

We agree with the reviewer, and more complex morphological information could certainly lead to exciting new insights in the future. To address the reviewer's question in a simple approach, we split the dataset into binucleated, mononucleated and no-nucleus shapes (Figure R1), and plotted the intensities of the four histone subunits shown in Figure 2B. This confirmed the visual information shown in the figure, i.e., that the lowest protein intensities corresponded to shapes without a visual nucleus. We now included the panel as supplementary Figure S3h.

Figure R1 and Supplementary Figure S3h, Levels of four histone proteins by number of nuclei. The number of nuclei was determined visually on corresponding single shape images.

- The GitHub reviewer access still asks for email confirmation, so it seems this approach of sharing the code in advance does not work.

We are sorry about this – it turns out that the IP addresses of all our machines that we tested are known to access the Mann labs GitHub account and therefore they did not require additional verification. Next time, we will make the GitHub code available to reviewers in a different style.

Reviewer #3:

Remarks to the Author:

The authors have extensively and adequately responded to my comments, as well as to the other reviewers. It is now a very nice piece of work

We thank the reviewer for supporting the manuscript in its current stage, and the comments of the first round

Final Decision Letter:

Dear Matthias,

I am pleased to inform you that your Article, "Spatial single-cell mass spectrometry defines zonation of the hepatocyte proteome", has now been accepted for publication in Nature Methods. Your paper is tentatively scheduled for publication in our October print issue, and will be published online prior to that. The received and accepted dates will be December 9, 2022 and August 15, 2023. This note is intended to let you know what to expect from us over the next month or so, and to let you know where to address any further questions.

Over the next few weeks, your paper will be copyedited to ensure that it conforms to Nature Methods style. Once your paper is typeset, you will receive an email with a link to choose the appropriate publishing options for your paper and our Author Services team will be in touch regarding any additional information that may be required.

You will receive a link to your electronic proof via email with a request to make any corrections within 48 hours. If, when you receive your proof, you cannot meet this deadline, please inform us at rjsproduction@springernature.com immediately.

Please note that *Nature Methods* is a Transformative Journal (TJ). Authors may publish their research with us through the traditional subscription access route or make their paper immediately open access through payment of an article-processing charge (APC). Authors will not be required to make a final decision about access to their article until it has been accepted. [Find out more about Transformative Journals](https://www.springernature.com/gp/open-research/transformative-journals)

Authors may need to take specific actions to achieve [compliance](https://www.springernature.com/gp/open-research/funding/policy-compliance-faqs) with funder and institutional open access mandates. If your research is supported by a funder that requires immediate open access (e.g. according to [Plan S principles](https://www.springernature.com/gp/open-research/plan-s-compliance)) then you should select the gold OA route, and we will direct you to the compliant route where possible. For authors selecting the subscription publication route, the journal's standard licensing terms will need

to be accepted, including [self-archiving policies](https://www.springernature.com/gp/open-research/policies/journal-policies). Those licensing terms will supersede any other terms that the author or any third party may assert apply to any version of the manuscript.

Your paper will now be copyedited to ensure that it conforms to Nature Methods style. Once proofs are generated, they will be sent to you electronically and you will be asked to send a corrected version within 24 hours. It is extremely important that you let us know now whether you will be difficult to contact over the next month. If this is the case, we ask that you send us the contact information (email, phone and fax) of someone who will be able to check the proofs and deal with any last-minute problems.

If, when you receive your proof, you cannot meet the deadline, please inform us at rjsproduction@springernature.com immediately.

Once your manuscript is typeset and you have completed the appropriate grant of rights, you will receive a link to your electronic proof via email with a request to make any corrections within 48 hours. If, when you receive your proof, you cannot meet this deadline, please inform us at rjsproduction@springernature.com immediately.

Once your paper has been scheduled for online publication, the Nature press office will be in touch to confirm the details.

Once your paper has been scheduled for online publication, the Nature press office will be in touch to confirm the details.

Content is published online weekly on Mondays and Thursdays, and the embargo is set at 16:00 London time (GMT)/11:00 am US Eastern time (EST) on the day of publication. If you need to know the exact publication date or when the news embargo will be lifted, please contact our press office after you have submitted your proof corrections. Now is the time to inform your Public Relations or Press Office about your paper, as they might be interested in promoting its publication. This will allow them time to prepare an accurate and satisfactory press release. Include your manuscript tracking number NMETH-A51109B and the name of the journal, which they will need when they contact our office.

About one week before your paper is published online, we shall be distributing a press release to news organizations worldwide, which may include details of your work. We are happy for your institution or funding agency to prepare its own press release, but it must mention the embargo date and Nature Methods. Our Press Office will contact you closer to the time of publication, but if you or your Press Office have any inquiries in the meantime, please contact press@nature.com.

Nature Portfolio journals [encourage authors to share their step-by-step experimental protocols](https://www.nature.com/nature-research/editorial-policies/reporting-standards#protocols) on a protocol sharing platform of their choice. Nature Portfolio 's Protocol Exchange is a free-to-use and open resource for protocols; protocols deposited in Protocol Exchange are citable and can be linked from the published article. More details can found at www.nature.com/protocolexchange/about.

Best regards,
Arunima

Arunima Singh, Ph.D.
Senior Editor
Nature Methods